# Learning Expressive Meta-Representations with Mixture of Expert Neural Processes

**Qi Wang**
Amsterdam Machine Learning Lab
University of Amsterdam
`hhq123go@gmail.com`

**Herke van Hoof**
Amsterdam Machine Learning Lab
University of Amsterdam
`h.c.vanhoof@uva.nl`

## Abstract

Neural processes (NPs) formulate exchangeable stochastic processes and are promising models for meta learning that do not require gradient updates during the testing phase. However, most NP variants place a strong emphasis on a global latent variable. This weakens the approximation power and restricts the scope of applications using NP variants, especially when data generative processes are complicated. To resolve these issues, we propose to combine the **M**ixture **o**f **E**xpert models with **N**eural **P**rocesse**s** to develop more expressive exchangeable stochastic processes, referred to as Mixture of Expert Neural Processes (MoE-NPs). Then we apply MoE-NPs to both few-shot supervised learning and meta reinforcement learning tasks. Empirical results demonstrate MoE-NPs' strong generalization capability to unseen tasks in these benchmarks.

## 1 Introduction

Humans can naturally accommodate themselves to new environments after developing related skills, and this kind of adaptability relies on the good abstraction of environments. Similarly, *meta learning* or *learning to learn* tries to leverage past experiences, and with help of the incorporated meta learned knowledge, a new skill can be mastered rapidly with a few instances.

During the past decade, an increasing number of methods have emerged in meta learning domains. In this paper, we concentrate on a special branch of meta learning methods, referred to as context-based meta learning [1; 2]. A representative one is a neural process (NP) [1], which was initially proposed to approximate Gaussian processes with lower computational cost. The core purpose of NPs is to learn meta-representations [3], which encode context points into latent variables and represent the task in a functional form. In comparison to gradient-based meta learning algorithms, *e.g.* model-agnostic meta learning (MAML) [4], the NP directly learns a functional representation and does not require additional gradient updates in fast adaptation.

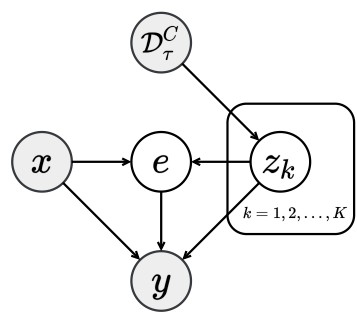

Figure 1: Generative Process of MoE-NPs. Here $\mathcal{D}_\tau^C$ refers to dataset of context points in the paper. $\{z_k\}_{k=1}^K$ are a set of expert latent variables and $e$ is an assignment latent variable. Observed variables are grey in circles with latent variables white.

**Research Motivations.** Fundamentally, vanilla NPs employ global Gaussian latent variables to specify different tasks. This setting raises several concerns in some scenarios. (i) When observations originate from a mixture of stochastic processes [5], a single Gaussian latent variable is faced with deficiencies in formulating complex functional forms. (ii) In context-based meta reinforcement learning, the uncertainty of value functions revealed from latent variables encourages effective exploration in environments [6]. But when tasks are governed by

36th Conference on Neural Information Processing Systems (NeurIPS 2022).

multiple variate, *e.g.* velocities, masses or goals in Mujoco robots [7], the use of a unimodal Gaussian latent variable restricts the flexibility of randomized value functions [8], leading to sub-optimality in performance.

**Developed Methods.** Instead of using a global latent variable in modeling, we employ multiple latent variables to induce a mixture of expert NPs to specify diverse functional priors. Meanwhile, the discrete latent variables as assignment variables are introduced to establish connections between a single data point and expert NPs. To optimize this model with hybrid types of latent variables, we utilize variational inference to formulate the evidence lower bound. Additionally, special modules are designed to accommodate few-shot supervised learning and meta reinforcement learning tasks.

**Outline & Contributions.** We overview the properties of NPs and general notations for context-based meta learning tasks in Section 3. Section 2 summarizes related work in meta learning and the NP family. In Section 4, **M**ixture **o**f **E**xpert **N**eural **P**rocesses (MoE-NPs) are elaborated, together with required modules for meta learning tasks. Experimental results and analysis are reported in Section 5. Discussions and limitations are included in Conclusion Section. In principle, our contribution is two-fold:

- We introduce a new exchangeable stochastic process, referred to as MoE-NPs, to enrich the family of NPs. Our model inherits the advantages of both mixtures of experts models and NPs, enabling multi-modal meta representations for functions.
- We specify inference modules in MoE-NPs for few shot supervised learning and meta reinforcement learning tasks. Extensive meta learning experiments show that MoE-NPs can achieve competitive performance in comparison to most existing methods.

## 2   Literature Review

**Meta Learning.** Meta learning is a paradigm to enable fast learning (fast adaptation to new tasks) via slow learning (meta training in the distribution of tasks). There exist several branches of meta learning algorithms. Gradient-based meta learning algorithms, *e.g.* MAML [4] and its variants [9; 10; 11], perform gradient updates over model parameters to achieve fast adaptation with a few instances. Metrics-based meta learning algorithms try to learn representations of tasks in distance space, and models *e.g.* prototypical networks [12; 13] are popular in computer vision domains. As for context-based meta learning methods of our interest, latent variable models, *e.g.* NPs [1], are designed to learn task representations in a functional space. This family does not require gradient updates in fast adaptation.

**Neural Processes Family.** Apart from vanilla NPs or CNPs [2; 1], other variants are developed and these are built on various inductive biases. To address underfitting issues, attention networks [14; 15] are introduced to augment NPs. [16; 17] improve the generalization capability of NPs with help of convolutional operations. To learn distributions of group equivariant functions, [18] has proposed EquiCNP. Similarly, SteerCNPs also incorporate equivariance to approximate stochastic fields [19]. Our work is to get NPs married with Mixture of Experts (MoEs) models [20; 21], which model datasets with a collection of expert NPs. We provide more technical summary of the NP family together with other probabilistic meta learning methods [22; 23; 24; 25] in the Appendix (F).

## 3   Preliminaries

**Notations.** The paradigm of *meta learning* is considered in the distribution of tasks $p(\mathcal{T})$, and a task sampled from $p(\mathcal{T})$ is denoted by $\tau$ in this paper. The form of a task depends on applications. For example, a task of our interest in regressions can be a function $f$ to fit, which is a realization from unknown stochastic processes [26].

Let $\mathcal{D}_\tau^C$ refer to a set of context points used to specify the underlying task, and $\mathcal{D}_\tau^T = [x_T, y_T]$ is a set of target points to predict, *e.g.* $f(x_T) = y_T$. In the context-based meta learning with latent variables, we write the probabilistic dependencies in distributions of target functions as follows,

$$p(f(x_T)|\mathcal{D}_\tau^C, x_T) = \int p(f(x_T)|z, x_T)p(z|\mathcal{D}_\tau^C)dz \tag{1}$$

where the functional prior $p(z|\mathcal{D}_\tau^C)$ is injected in modeling via latent variables $z$.

**Neural Processes.** The family of NPs [1] belongs to exchangeable stochastic processes [27]. A generative process is written as Eq. (2) with a global Gaussian latent variable $z$ placed in Eq. (1),

$$\rho_{x_{1:N}}(y_{1:N}) = \int p(z) \prod_{i=1}^{N} \mathcal{N}(y_i | f_\theta(x_i, z), \sigma_i^2) dz \tag{2}$$

where $f_\theta$ is a mean function and $\sigma_i^2$ is the corresponding variance. In our settings, we treat the conditional neural process [2] as a special case in NPs, when the distribution of $z$ is collapsed into a Dirac delta distribution $p(z) = \delta(z - \hat{z})$ with $\hat{z}$ a fixed real valued vector.

### 3.1 Few-Shot Supervised Learning

In the context-based meta learning, we formulate the few-shot supervised learning objective within the expected risk minimization principle as follows.

$$\min_{\Theta} \mathbb{E}_{\tau \sim p(\mathcal{T})} \big[ \mathcal{L}(\mathcal{D}_\tau^T; \mathcal{D}_\tau^C, \Theta) \big] \tag{3}$$

The risk function $\mathcal{L}$, *e.g.* negative log-likelihoods, measures performance of meta learning structure on the task-specific dataset $\mathcal{D}_\tau^C$ and $\mathcal{D}_\tau^T$, and $\Theta$ means parameters of common knowledge shared across tasks and parameters for fast adaptation (*e.g.* $\Theta$ denotes parameters of the encoder $\phi$ and decoder $\theta$, and $\mathcal{L}$ is the approximate objective in NPs).

With the set of context points $\mathcal{D}_\tau^C = \{(x_1, y_1), \ldots, (x_m, y_m)\}$ and the target points $\mathcal{D}_\tau^T$, the posterior of a global latent variable $z$ in Eq. (1) is approximated with a variational distribution $q_\phi(z | \mathcal{D}_\tau^T)$ and an evidence lower bound (ELBO) is derived to optimize in practice. A general meta training objective of NPs in few-shot supervised learning is $\mathcal{L}(\theta, \phi)$ in Eq. (4), where $p_\theta(\mathcal{D}_\tau^T | z) = \prod_{i=1}^{n} p_\theta(y_i | [x_i, z])$.

$$\mathbb{E}_\tau \big[ \ln p(\mathcal{D}_\tau^T | \mathcal{D}_\tau^C) \big] \geq \mathbb{E}_\tau \big[ \mathbb{E}_{q_\phi(z | \mathcal{D}_\tau^T)} [\ln p_\theta(\mathcal{D}_\tau^T | z)] - D_{KL}[q_\phi(z | \mathcal{D}_\tau^T) \| q_\phi(z | \mathcal{D}_\tau^C)] \big] \tag{4}$$

### 3.2 Meta Reinforcement Learning

For the context-based meta reinforcement learning, the context points $\mathcal{D}_\tau^C$ are a set of random transition samples from an environment as $\mathcal{D}_\tau^C = \{(s_1, a_1, s_2, r_1), \ldots, (s_H, a_H, s_{H+1}, r_H)\}$, where $r_t$ is the one-step reward after performing action $a_t$ at state $s_t$. Here $\mathcal{D}_\tau^C$ plays a role in task inference [28] to obtain the information bottleneck $q_\phi(z | \mathcal{D}_\tau^C)$ and $\mathcal{D}_\tau^T$ is the dataset of state action values to fit.

For example, in an off-policy meta reinforcement learning algorithm, *e.g.* PEARL [6] or FCRL [3], the general optimization objective consists of two parts: (i) to approximate distributions of task-specific optimal value functions in Eq. (5), where $Q_\theta$ is optimal $Q$-value with the state value $\hat{V}$ (ii) to maximize the cumulative rewards $\mathbb{E}_\tau \big[ \mathbb{E}_{q_\phi(z | \mathcal{D}_\tau^C)} [\mathcal{R}(\tau, z; \varphi)] \big]$, where $\mathcal{R}$ is the expected cumulative rewards in the environment $\tau$ given policies $\pi_\varphi(a | [s, z])$.

$$\mathcal{L}(\theta, \phi) = \mathbb{E}_\tau \mathbb{E}_{\substack{(s, a, s', r) \sim \mathcal{D}_\tau^T \\ z \sim q_\phi(z | \mathcal{D}_\tau^C)}} [Q_\theta([s, z], a) - (r + \hat{V}([s', z]))]^2 + \beta \mathbb{E}_\tau \big[ D_{KL}[q_\phi(z | \mathcal{D}_\tau^C) \| p(z)] \big] \tag{5}$$

Different from the few-shot supervised learning, here the context points are not part of fitting dataset, which means $\mathcal{D}_\tau^C \not\subset \mathcal{D}_\tau^T$. As implemented in [6], the prior distribution $p(z)$ is typically selected as a fixed one, *e.g.* $\mathcal{N}(0, I)$. The induced distribution of task-specific value functions $p(Q_\theta([s, z], a))$ enables posterior sampling [29] in meta learning scenarios, which brings additional benefits of exploration for continuous control problems.

## 4 Model & Algorithm

In this section, we present our developed MoE-NPs and connect them to the hierarchical Bayes framework. Then approximate objectives are derived and stochastic gradient variational Bayes [30] is used to optimize the developed model. Finally, specialized neural modules are described for

MoE-NPs application to different meta learning tasks. We have attached concepts of variational priors and posteriors and detailed computational diagrams in training and testing in Appendix (C). For the sake of simplicity, we derive equations *w.r.t.* a task $\tau$ in the following section, but a batch of tasks are considered in training in implementation.

## 4.1 Mixture of Expert Neural Processes

Vanilla NPs often suffer underfitting in experiments [1; 14]. This can be attributed to expressiveness bottlenecks when employing a global latent variable in learning functional priors of tasks from unknown distributions [31].

To alleviate mentioned deficiencies, we make two modifications for the NP family. (i) Multiple functional priors are encoded in modeling with help of $K$ expert latent variables, which can capture statistical traits, *e.g.* distributional multi-modality, in data points. This setting is also an extension of Mixture of Experts (MoEs) models [20; 32; 33; 34] to meta learning scenarios. (ii) Like the gating mechanism in [5], assignment latent variables are included in modeling to select functional forms for each data point in prediction. The resulting MoE-NPs can learn more expressive functional priors and exhibit the approximation power for local properties of the dataset.

**Generative Process.** As displayed in Fig. (1), the graphical model involves two types of latent variables, respectively the continuous expert latent variables $z_{1:K}$ and the discrete assignment latent variable $e$. Further, we can translate the generative process into equations as follows,

$$\rho_{x_{1:N}}(y_{1:N}) = \int \prod_{k=1}^{K} p(z_k) \cdot \prod_{i=1}^{N} \left[ \sum_{k=1}^{K} p(y_i|x_i, z_{1:K}, e_k = 1)p(e_k = 1|x_i, z_{1:K}) \right] dz_{1:K} \quad (6)$$

where the sampled assignment variable $e$ is in the form of $K$-dimensional one-hot encoding, and $e_k = 1$ in Eq. (6) indicates the $k$-th expert $z_k$ is selected from $z_{1:K}$ for prediction. A more detailed probabilistic generative process can also be found in Appendix (E.1). In this way, our developed model constitutes an *exchangeable stochastic process*. And we demonstrate this claim with help of Kolmogorov Extension Theorem [35] in Appendix (E.2).

**Link to Hierarchical Bayes.** Note that latent variables in Eq. (6) are of hybrid types. $K$ functional priors are incorporated in expert latent variables $z_{1:K}$, while the assignment latent variable $e$ is input dependent. The dependencies between $z_{1:K}$ and $e$ are reflected in modeling, and this connects our work to Hierarchical Bayes [36; 37] in a latent variable sense. Also when only one expert latent variable is used here, the hierarchical model degenerates to the vanilla (C)NPs [1; 2].

## 4.2 Scalable Training & Prediction

**Inference Process.** Given a task $\tau$, due to existence of unknown latent variables, it is intractable to perform exact inference *w.r.t.* $p(\mathcal{D}_\tau^T|\mathcal{D}_\tau^C)$. As an alternative, we apply variational inference to our developed model. Here we use $[x, y]$ to denote a single data point from a set of target points $\mathcal{D}_\tau^T$.

$$\ln p(y|x, \mathcal{D}_\tau^C) \geq \mathbb{E}_{q_{\phi_1}} \left[ \mathbb{E}_{q_{\phi_{2,1}}} [\ln p_\theta(y|x, z_{1:K}, e)] - D_{KL}[q_{\phi_{2,1}}(e|x, y, z_{1:K}) \| p_{\phi_{2,2}}(e|x, z_{1:K})] \right]$$

$$- \sum_{k=1}^{K} D_{KL}[q_{\phi_{1,k}}(z_k|\mathcal{D}_\tau^T) \| q_{\phi_{1,k}}(z_k|\mathcal{D}_\tau^C)] = -\mathcal{L}(\theta, \phi_1, \phi_2)$$

$$(7)$$

An example for the $k$-th expert latent variable $z_k$ can be in the form of a Gaussian distribution $\mathcal{N}(z_k; \mu_k, \Sigma_k)$. And in meta training processes, the assignment variable $e$ is assumed to be drawn from a categorical distribution $\mathtt{Cat}(e; K, \alpha(x, y, z_{1:K}))$ with parameters $\alpha(x, y, z_{1:K})$. The existence of discrete latent variables $e$ makes it tough to optimize using traditional methods. This is because either sampling algorithms or expectation maximization algorithms are computationally intensive when utilized here (we have discussed this point in Appendix (G)) for expert latent variables. To reduce computational cost, we again utilize variational inference and the decoder directly formulates the output as a mixture of log-likelihoods $\mathbb{E}_{q_{\phi_{2,1}}} [\ln p_\theta(y|x, z_{1:K}, e)] = \sum_{k=1}^{K} \alpha_k(x, y, z_{1:K}; \phi_{2,1}) \ln p_\theta(y|x, z_k)$.

This results in a general ELBO as Eq. (7) for few-shot supervised learning in meta training, where $q_{\phi_1}$ denotes a collection of $K$ independent variational distribution $\{q_{\phi_{1,1}}, \ldots, q_{\phi_{1,K}}\}$. $q_{\phi_{2,1}}$ and $p_{\phi_{2,2}}$ respectively define the variational posterior and prior for assignment latent variables. Please refer to Appendix (C)/(G) for definitions and more detailed derivations.

**Monte Carlo Estimates & Predictions.** Meta-training processes consider a batch of tasks to optimize in iterations, and we apply Monte Carlo methods to the obtained negative ELBO $\mathcal{L}(\theta, \phi_1, \phi_2)$ as follows.

$$\mathcal{L}_{MC}(\theta, \phi_1, \phi_2) = -\frac{1}{NB} \sum_{b=1}^{B} \sum_{i=1}^{N} \left[ \sum_{k=1}^{K} \alpha_k^{(b)} \ln p(y_i^{(b)}|x_i^{(b)}, z_k^{(b)}) \right]$$

$$+ \frac{1}{NB} \sum_{b=1}^{B} \sum_{i=1}^{N} D_{KL}[q_{\phi_{2,1}}(e_i^{(b)}|x_i^{(b)}, y_i^{(b)}, z_{1:K}^{(b)}) \parallel p_{\phi_{2,2}}(e_i^{(b)}|x^{(b)}, z_{1:K}^{(b)})] \qquad (8)$$

$$+ \frac{1}{NB} \sum_{b=1}^{B} \sum_{k=1}^{K} D_{KL}[q_{\phi_{1,k}}(z_k^{(b)}|\mathcal{D}_b^T) \parallel q_{\phi_{1,k}}(z_k^{(b)}|\mathcal{D}_b^C)]$$

With the number of tasks $B$ and the number of data points $N$ in mini-batches, the Monte Carlo estimate with one stochastic forward pass is Eq. (8) for meta training objectives.

Like that in NPs [1], we derive the predictive distribution as Eq. (9) with one stochastic forward pass and parameters of discrete latent variables $p_{\phi_{2,2}}(e_k = 1|x_*, z_{1:K}) = \alpha_k(x, z_{1:K}; \phi_{2,2})$.

$$p(y_*|x_*, \mathcal{D}_\tau^C) \approx \sum_{k=1}^{K} \alpha_k(x, z_{1:K}; \phi_{2,2}) p_\theta(y|x_*, z_k) \quad \text{with } z_{1:K} \sim q_{\phi_1}(z_{1:K}|\mathcal{D}_\tau^C) \qquad (9)$$

And the point estimate in prediction $\mathbb{E}[Y|X = x, \mathcal{D}_\tau^C]$ can also be obtained in Appendix (G.4).

### 4.3 Module Details for Meta Learning

#### 4.3.1 Inference Modules in MoE-NPs

The equations so far define a framework that can be implemented in different meta learning tasks. Two examples are given in Fig. (2). Note that two types of latent variables are involved in modelling, we need different structures of encoders for latent variables. Inference Modules are required for them, satisfying different conditions. In variational inference, distribution parameters of these latent variables are approximated with the output of these specialized encoders.

**Inference Modules for Continuous Latent Variables.** For neural networks to parameterize the encoder of continuous latent variables $q_{\phi_1}$, we use the same architectures in (C)NPs [1; 2], which are permutation invariant to the order of context points $[x_C, y_C] = \{[x_1, y_1], \ldots, [x_N, y_N]\}$. That is, for any permutation operator $\sigma$ over the set of context points, the neural network (NN) parameters of an output distribution for each expert $z_k$ should satisfy $[\mu_k, \Sigma_k] = \text{NN}_{\phi_{1,k}}([x_{\sigma(1:N)}, y_{\sigma(1:N)}])$.

$$r_{k,i} = h_{\phi_{1,k}}([x_i, y_i]), \quad r_k = \bigoplus_{i=1}^{N} r_{k,i}, \quad [\mu_k, \Sigma_k] = g_{\phi_{1,k}}(r_k) \qquad (10)$$

Eq. (10) is an example, where $h$ is the embedding function, $\bigoplus$ denotes a mean pooling operation, and $g$ is part of encoder networks.

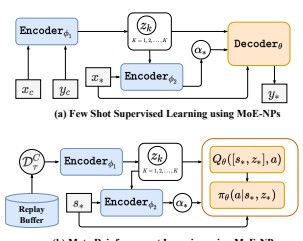

(a) Few Shot Supervised Learning using MoE-NPs

(b) Meta Reinforcement Learning using MoE-NPs

Figure 2: Computational Diagram of MoE-NPs in Meta Testing. **In (a):** The context variables are $\mathcal{D}_\tau^C = [x_C, y_C]$, and the expert latent variables $z_{1:K}$ are approximated with neural networks. For discrete assignment latent variables $e_*$, we learn parameters of categorical distributions $\alpha_*$ with neural networks. **In (b):** The context variables $\mathcal{D}_\tau^C$ are sampled transitions from a memory buffer. The selected expert latent variable $z_*$ is a context variable in both Actor and Critic networks during policy search.

**Inference Modules for Categorical Latent Variables.** For neural networks to parameterize the encoder of discrete latent variables $q_{\phi_{2,1}}$ and $p_{\phi_{2,2}}$, we need the categorical distribution parameters $\alpha$

to be permutation equivariant [38] with respect to the order of $z_{1:K}$. This means for any order permutation $\sigma$, the condition is satisfied as $[\alpha_{\sigma(1)}, \alpha_{\sigma(2)}, \ldots, \alpha_{\sigma(K)}] = \texttt{NN}_{\phi_{2,1}}(x, y, z_{\sigma(1:K)})$.

$$b_k = h_{\phi_{2,1}}(x, y, z_k) \, \forall k \in \{1, 2, \ldots, K\}, \quad [\alpha_1, \alpha_2, \ldots, \alpha_K] = \texttt{softmax}(b/t) \tag{11}$$

An example implementation for the variational posterior $\texttt{Cat}(e; K, \alpha(x, y, z_{1:K}))$ can be Eq. (11), where the vector of logits is $b = [b_1, b_2, \ldots, b_K]$ with $t$ a temperature parameter. And this implementation applies to prior networks $\texttt{NN}_{\phi_{2,2}}(x, z_{\sigma(1:K)})$ to learn distribution parameters of $\texttt{Cat}(e; K, \alpha(x, z_{1:K}))$ in the same way.

### 4.3.2 Meta RL Modules in MoE-NPs.

When extending MoE-NPs to meta RL tasks, optimization objectives in Eq. (7) need to be modified for Actor-Critic methods, which are employed in our settings. Like that in PEARL [6] and FCRL [3], the soft actor critic (SAC) algorithm [39] is used to learn policies due to good sample efficiency.

Given a specific MDP $\tau$, posterior distributions of optimal value functions are formulated via latent variables $z$ in context-based meta RL. That is, $p(Q_\theta(s, a; \mathcal{M}))$ is approximated in the form $p(Q_\theta([s, z], a))$. The resulting objectives for the Actor and Critic functions are respectively in Eq. (12) and Eq. (13), where $\mathcal{Z}_\theta$ is a normalization factor.

$$\mathcal{L}_A^\tau = \mathbb{E}_{\substack{s \sim \mathcal{D}_\tau^T, a \sim \pi_\varphi \\ z \sim q_\phi}} \left[ D_{KL} \left[ \pi_\varphi(a|[s, z]) \, \| \, \frac{\exp\left(Q_\theta([s, z], a)\right)}{\mathcal{Z}_\theta(s)} \right] \right] \tag{12}$$

The variational posterior $q_\phi(z|s, \mathcal{D}_\tau^C)$ in Eq. (13) is a state dependent distribution with $\hat{V}$ a state value function, and sampling processes refer to steps in Algorithm (3).

$$\mathcal{L}_C^\tau = \mathbb{E}_{\substack{(s,a,s',r) \sim \mathcal{D}_\tau^T \\ z, z' \sim q_\phi}} [Q_\theta([s, z], a) - (r + \hat{V}([s', z']))]^2 \tag{13}$$

A key difference from PEARL [6] lies in that several expert latent variables and assignment latent variables are involved in modeling. So we refer the Kullback–Leibler divergence term to Eq. (14) in MoE-NPs with coefficient $\beta_0$ and $\beta_1$.

$$\mathcal{L}_{KL}^\tau = \beta_1 \mathbb{E}_{\substack{(s,a,s',r) \sim \mathcal{D}_\tau^T \\ q_{\phi_1}(z_{1:K}|\mathcal{D}_\tau^C)}} [D_{KL}[q_{\phi_2}(e|s, z_{1:K}) \, \| \, p(e)]] + \beta_0 \sum_{k=1}^K D_{KL}[q_{\phi_{1,k}}(z_k|\mathcal{D}_\tau^C) \, \| \, p(z_k)] \tag{14}$$

The Monte Carlo estimates *w.r.t.* Eq. (12/13/14) are used in meta training, and Pseudo code to optimize these functions is listed in Appendix (A).

## 5 Experiments and Analysis

### 5.1 General Settings

The implementation of MoE-NPs in meta training can be found in Appendix Algorithms (1)/(3), and also please refer to Appendix Algorithms (2)/(4) for the corresponding meta-testing processes. We leave the details of experimental implementations (*e.g.* parameters, neural architectures, corresponding PyTorch modules and example codes) in Appendix (H).

**Baselines for Learning Tasks.** Apart from MoE-NPs, methods involved in comparisons are context-based methods such as CNPs [2], NPs [1] and FCRL [3], and gradient-based methods such as MAML [4] and CAVIA [9]. For FCRL, contrastive terms from SimCLR [40] are included in the objective. In meta RL, the modified NP model corresponds to PEARL [6]. Meanwhile, in Appendix (I), we include additional experimental results compared with other NPs models augmented by attentive modules [14] or convolutional modules [17].

## 5.2 Illustration in Toy Regression

To see different roles of latent variables, we visually show effects of stochastic function fitting and quantified uncertainty in toy dataset. Our goal is to discover potential components of distributions from limited observed data points.

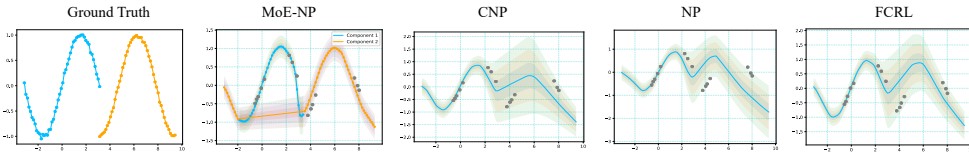

Figure 3: The Ground Truth and Predictive Distributions of Curves using NP related Models. The gray dots around curves are the context points. The shaded regions correspond to $3x$ standard deviations. In MoE-NPs, two components of the sampled mixture curve in blue and orange can be identified via assignment latent variables with more than 85% accuracy.

The learning data points are sampled in $x$-domain $[-\pi, 3\pi]$ and merged from a mixture of randomized functions $f_1(x) = \sin(x) + \epsilon_1$ and $f_2(x) = \cos(x) + \epsilon_2$ with equal probability for mixture components, where $\epsilon_1 \sim \mathcal{N}(0, 0.03^2)$ and $\epsilon_2 \sim \mathcal{N}(0, 0.01^2)$. In each training iteration, we sample a batch of data points and randomly partition context points and target points for learning. In testing phase, we draw up 100 data points from this mixture of distributions with 15 random data points selected as the context. The fitting results for one sampled mixture curve are shown in Fig. (3). It can be seen that both CNPs and FCRL display similar patterns, overestimate the uncertain in the mixture curve of the second component. NPs show intermediate performance and still fails to match context points well. As for MoE-NPs, with help of predicted assignment variables parameters $e_* = \texttt{one\_hot}[\arg_{k \in \{1,2\}} \max \alpha_k]$, we set the number of experts as two and partition data points to visualize predictive distributions $p_\theta(y_*|x_*, z_*)$ . The MoE-NP is able to precisely separate mixture components inside the dataset and provides more reliable uncertainty.

## 5.3 Few-Shot Supervised Learning

We evaluate the performance of models on a system identification task in Acrobot [41] and image completion task in CIFAR10 [42]. Both tasks are common benchmarks in the meta learning or NPs literature [41; 43; 1; 2; 14].

Table 1: System Identification Performance in Meta Testing Acrobot Tasks. Shown are mean square errors and standard deviations in fitting meta-testing tasks. Figures in the Table are scaled by multiplying E-3 for means and standard deviations. The best results are in bold.

| CNP | NP | FCRL | MAML | CAVIA | MoE-NP |
|---|---|---|---|---|---|
| 2.3($\pm$0.13) | 7.2($\pm$0.5) | 2.0($\pm$0.15) | 2.5($\pm$0.35) | 2.0($\pm$0.23) | **1.4($\pm$0.06)** |

**System Identification.** For Acrobot systems, different tasks are generated by varying masses of two pendulums. A dataset of state transitions is collected by using a complete random policy to interact with sampled environments. The state consists of continuous as angles and angular velocities $[\theta_1, \theta_1', \theta_2, \theta_2']$. The objective is to predict the resulting state after a selected Torque action from $\{-1, 0, +1\}$. For more details about meta training dataset and environment information, refer to Appendix (H.2).

In the meta testing phase, 15 episodes with the length of horizon 200 are collected for each task and we report the average predictive errors and standard deviations for all transitions. Here we use 50 transitions as the context points to identify the task. As exhibited in Table (1), gradient-based methods, *e.g.* CAVIA and MAML, beat NP in terms of predictive accuracy but show higher variances than all other models. With three experts in modeling, MoE-NPs significantly outperform other baselines in terms of dynamics prediction. Our finding is consistent with observations in [44], where multi-modal distributions are necessary for Acrobot systems. We also illustrate the asymptotic performance of MoE-NPs with the increase of the context points in the following Section (5.5) Ablation part.

**Image Completion.** We use CIFAR10 dataset [42] in this experiment, which is formulated with 32x32 RGB images. In the meta training process, a random number of pixels are masked to complete

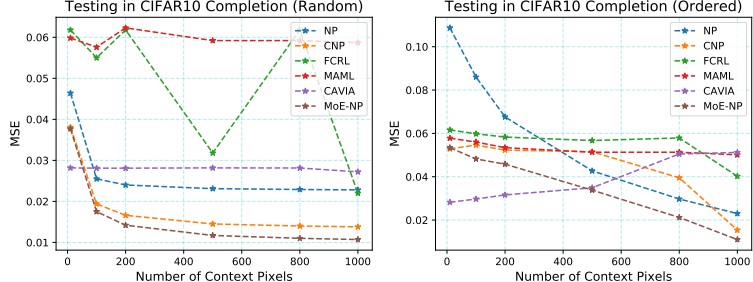

Figure 4: CIFAR10 Completion Performance with Various Number of Context Pixels. The numbers of context points used in prediction are 10, 100, 200, 500, 800, 1000. The left figure is with random context pixels while the right one is with the ordered context pixels.

in images. That is, given the context pixel locations and values $[x_C, y_C]$, we need to learn a map from each 2-D pixel location $x \in [0, 1]^2$ to pixel values $y \in \mathbb{R}^3$. Here two expert latent variables are used in MoE-NPs.

In Fig. (4), we evaluate image completion performance on the test dataset and the number of context pixels is varied in three levels. It can be found that CAVIA works best in cases with 10 random context pixels or less than 500 ordered context pixels. In other cases, MoE-NP surpasses all baselines. With more observed pixels, the predictive errors of MoE-NPs can be decreased in both random and ordered context cases. An example of image completion results is displayed in Fig. (5).

For gradient-based methods, CAVIA and MAML are sensitive to the number of context points and do not exhibit asymptotic performance like that in MoE-NP. NP still suffers underfitting in performance.

## 5.4 Meta Reinforcement Learning

To evaluate the meta RL implementation of our model, we conduct the experiments in a 2-D point robot and Mujoco environments [7]. Fig. (6) exhibits the environments used in this paper, and we leave more details in Appendix (H.1).

**2D Navigation.** For 2-D point robot tasks, the agent needs to navigate with sparse rewards. The navigation goal of a task is sampled from a mixture of arcs in a semi-circle in Fig. (6.a) during meta training processes.

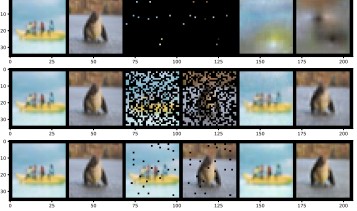

Figure 5: Image Completion Visualization using MoE-NPs. From the Top to the Bottom: the number of random context pixels are 10, 500 and 1000. From the Left to the Right (every two): original images, masked images and reconstructed images.

From Fig. (7.a-c), we can observe the evaluation performance of agents over iterations. For gradient-based methods, CAVIA shows better performance than MAML in exploration but both are weaker than context-based baselines. MoE-NPs can converge earlier with less training samples and show slight advantage over vanilla PEARL. In particular, we test the asymptotic performance in out of distributions (O.O.D.) tasks and show results in Fig. (7.d). We notice O.O.D. tasks are challenging for all algorithms to generalize but average returns are gradually increased with more trials. PEARL and FCRL achieve comparable rewards, while MoE-NP behaves better in this case.

**Locomotion.** As for Half Cheetah-Complex-Direction (H-Cheetah-CD) and Slim Humanoid-Complex-Goal (S-Humanoid-CG) tasks, these correspond to locomotion in complicated environments. Note that multiple directions and goals are involved in tasks.

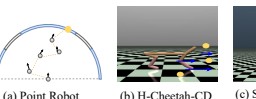

Figure 6: Environments for Meta Reinforcement Learning. **In (a)**: Blue arcs are distributions of goals in orange for the robot to reach with sparse rewards. **In (b)/(c)**: Goals in orange and directions in blue are varied in tasks.

Fig. (7) illustrates the performance of learned policies in meta learning tasks. In H-Cheetah-CD, MoE-NP shows a slight advantage over FCRL, and it exhibits comparable performance in S-Humanoid-CG. In both environments, MoE-NP and FCRL outperform other baselines. This implies the importance of func-

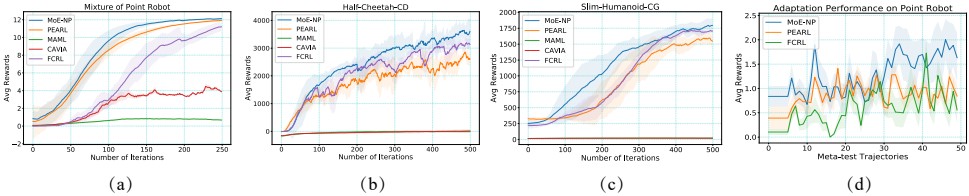

(a)        (b)        (c)        (d)

Figure 7: Results in Meta Learning Continuous Control. **In (a)/(b)/(c):** Learning curves show tested average returns with variances in 4 runs. For point robot environments, 100 transitions are randomly collected from a task specific memory buffer to infer the posterior. For Mujoco environments, 400 transitions are randomly collected from a task specific memory buffer to infer the posterior. **In (d):** Fast Adaptation Performance in Meta Testing Point Robot Environments. The collected episodes are gradually increased to 50 and the average returns together with variances are visualized. 5 goals are sampled from the white part of arcs in Fig. (6.a).

tional representations for task-specific value functions. Either contrastive or multiple functional priors lead to better exploration and have a potential to boost performance in continuous control. For gradient-based methods, observations show that they can easily get stuck in the local optimal [6; 45].

## 5.5 Ablation Studies

**Number of Experts.** We examine the influence of the number of experts in meta-trained MoE-NPs, and system identification in the Acrobot system is selected as an example here. As displayed in Fig. (8), the number of experts are 3, 5, 7 and 9 in different MoE-NPs. Here we test the predictive performance of meta-trained MoE-NP by varying the number of transitions. We set respectively 15, 25, 50, 100 transition samples as the number of context points to identify the system. All settings for meta testing processes are already described in Section (5.3). It can be seen when the expert number is 5, the predictive performance is largely enhanced with the increase of context points and the variance shrinks accordingly. But with more experts, *e.g.* greater than 5, MoE-NPs exhibit higher predictive errors, and show no significant performance improvement with the increase of the number of context points. These suggest increasing the number of experts beyond a certain point can deteriorate the predictive performance of the MoE-NPs.

**Latent Variables in Meta RL.** As mentioned in Preliminaries Section, the use of latent variable is able to induce task-specific optimal value functions. Here we take the S-Humanoid-CG as the example, and the expert encoder of MoE-NPs (Deterministic) is set to the deterministic. In Fig. (9), we observe the performance degrades a lot using deterministic expert latent variables. These further verify findings in PEARL [6]. The randomness of value function distributions captures task uncertainty and encourages more efficient exploration.

## 6 Conclusions

**Technical Discussions.** In this work, we have developed a new variant of NP models by introducing multiple expert latent variables. Our work illustrates the roles of different latent variables in MoE-NPs for meta learning tasks. MoE-NPs are able to separate data points from different clusters of stochastic processes and exhibit superior performance in few-shot supervised learning tasks. Also, MoE-NPs are consistently among the best methods in meta learning continuous control.

**Existing Limitations.** Though the developed model provides more expressive functional priors, the appropriate number of experts is still hard to determine. Also, the mechanism of gradually

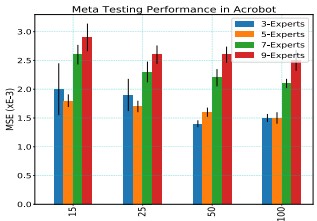

Figure 8: Predictive Performance of MoE-NPs in Acrobot Meta Testing Processes using Varying Numbers of Expert Latent Variables and Context Points. The scale for mean square errors together with standard deviations is E-3.

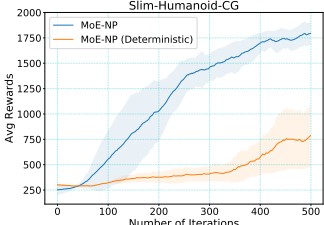

Figure 9: Ablation Performance in S-Humanoid-CG. Learning curves display tested average returns with variances in 4 runs.

incorporating new expert latent variables has not been explored and this raises concerns in additional computational cost and more effective inference.

**Future Extensions.** Here we provide a couple of heuristics to determine optimal number of experts for MoE-NPs in the future. Information metrics, *e.g.* Bayesian information criterion, can be incorporated in modeling. Another way is to place priors over distributions of discrete latent variables like that in hierarchical Dirichlet processes [46] and select the optimal number of experts in a Bayesian way.

## Acknowledgement

We thank NeurIPS anonymous reviewers and meta reviewers for their comments and constructive suggestions for our manuscript. Their engagement in the review/rebuttal period helps improve our manuscript a lot.

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
