# Contents

# A    Pseudo Code of Algorithms in Meta Learning

---

**Algorithm 1:** MoE-NPs for Few-Shot Supervised Learning.

**Input**  **:** Task distribution $p(\mathcal{T})$; Task batch size $\mathcal{B}$; Length of mini-batch instances $N_{max}$; Epochs $m$; Learning rates $\lambda_1$ and $\lambda_2$.

**Output :** Meta-trained parameters $\phi = [\phi_1, \phi_2]$ and $\theta$.

**1** Initialize parameters $\phi$ and $\theta$;

**2 for** $i = 1$ *to* $m$ **do**

**3**    Sample a random value $N_C \sim U[1, N_{max}]$

**4**    as the number of context points;

**5**    Sample mini-batch instances $\mathcal{D}$

**6**    to split dataset $\{(x_C, y_C, x_T, y_T)_{bs}\}_{bs=1}^{B}$;
    // generative process

**7**    Sample expert latent variables $z_{1:K} \sim q_{\phi_1}(z_{1:K}|\mathcal{D}^T)$;

**8**    Compute distribution parameters $\alpha$ with Eq. (11);

**9**    Compute negative ELBOs $\mathcal{L}_{MC}(\theta, \phi)$ in Eq. (8);
    // amortized inference process

**10**    $\phi \leftarrow \phi - \lambda_1 \nabla_\phi \mathcal{L}_{MC}(\theta, \phi)$ in Eq. (8);

**11**    $\theta \leftarrow \theta - \lambda_2 \nabla_\theta \mathcal{L}_{MC}(\theta, \phi)$ in Eq. (8);

**12 end**

---

**Algorithm 2:** MoE-NPs for Few Shot Supervised Learning (Meta-Testing Phases).

**Input**  **:** Task $\tau$; Meta-trained $\phi = [\phi_1, \phi_{2,2}]$ and $\theta$.

**Output :** Predictive distributions.

**1** Initialize parameters $\phi$ and $\theta$;

**2** Set the number of context points $N_C$;

**3** Split test dataset into the context/target

**4** $\{(x_C, y_C, x_T, y_T)_{bs}\}_{bs=1}^{B} \sim \mathcal{D}$;
  // generative process

**5** Sample expert latent variables of

**6** the mini-batch $z_{1:K} \sim q_{\phi_1}(z_{1:K}|\mathcal{D}_\tau^C)$;

**7 if** *discrete l.v.s for hard assignment* **then**

**8**    Sample assignment latent variables of

**9**    the mini-batch $e \sim p_{\phi_{2,2}}(e|x_T, z_{1:K})$;

**10**    Output the distribution $p_\theta(y_T|x_T, z_{1:K}, e)$;

**11 else**

**12**    Compute the distribution parameters $\alpha$

**13**    of assignment latent variables via Eq. (11);

**14**    Output the predictive distribution

**15**    $p_\theta(y_T|x_T, \mathcal{D}_\tau^C)$ via Eq. (9);

**16 end**

---

**Algorithm 3:** MoE-NPs for Meta RL.

**Input** : MDP distribution $p(\mathcal{T})$; Batch size of tasks $\mathcal{B}$; Training steps $m$; Learning rates $\lambda_1$, $\lambda_2$ and $\lambda_3$.

**Output :** Meta-trained parameters $\phi$, $\theta$ and $\varphi$.

1 Initialize parameters $\phi$, $\theta$, $\varphi$ and replay buffer$\{\mathcal{M}_\tau^C\}^{\mathcal{B}}$;

2 **while** *Meta-Training not Completed* **do**

3      Sample a batch of tasks $\{\tau\}^{\mathcal{B}} \sim p(\mathcal{T})$;

     // collect context transitions

4      **for** *each $\tau \in \{\tau\}^{\mathcal{B}}$* **do**

5          Initialize the context $\mathcal{D}_\tau^C = \{\}$;

6          Execute Algorithm (4) in Appendix

7          to update $\mathcal{D}_\tau^C$

8      **end**

     // actor critic learning in batchs

9      **for** $i = 1$ *to* $m$ **do**

10          **for** *each $\tau \in \{\tau\}^{\mathcal{B}}$* **do**

11              Sample context points $\mathcal{D}_\tau^C \sim \mathcal{S}_c(\mathcal{M}_\tau^C)$

12              & batch of transitions $b_\tau \sim \mathcal{M}_\tau$;

13              Sample $z_{1:K} \sim q_{\phi_1}(z_{1:K}|\mathcal{D}_\tau^C)$;

14              **for** *each $s \in b_\tau$* **do**

15                  Sample $e \sim q_{\phi_2}(e|z_{1:K}, s)$ to select $z$

16                  and augment the state as $[s, z] \in b_\tau$;

17              **end**

             // run forward propagation

18              $\mathcal{L}_A^\tau = \mathcal{L}_A^\tau(b_\tau)$ in Eq. (12);

19              $\mathcal{L}_C^\tau = \mathcal{L}_C^\tau(b_\tau)$ in Eq. (13);

20              $\mathcal{L}_{KL}^\tau = \mathcal{L}_{KL}^\tau(\mathcal{D}_\tau^C, b_\tau)$ in Eq. (14)

21          **end**

         // run back propagation

22          $\phi \leftarrow \phi - \lambda_1 \nabla_\phi \sum_\tau (\mathcal{L}_C^\tau + \mathcal{L}_{KL}^\tau)$ in Eq. (13/14);

23          $\varphi \leftarrow \varphi - \lambda_2 \nabla_\varphi \sum_\tau \mathcal{L}_A^\tau$ in Eq. (12);

24          $\theta \leftarrow \theta - \lambda_3 \nabla_\theta \sum_\tau \mathcal{L}_C^\tau$ in Eq. (13);

25      **end**

26 **end**

---

**Algorithm 4:** MoE-NPs for Meta RL (Meta-Testing Phases).

**Input** : MDP distribution $p(\mathcal{T})$; meta-trained parameters $\phi$, $\theta$.

**Output :** Cumulative rewards.

1 Sample a test task $\tau \sim p(\mathcal{T})$;

2 Initialize parameters $\phi$, $\theta$, $\varphi$ and replay buffer $\mathcal{M}_\tau^C$;

     // collect transitions for memory buffers

3 Initialize the context $\mathcal{D}_\tau^C = \{\}$;

4 **for** $k = 1, 2, \ldots, K$ **do**

5      Sample $z_{1:K} \sim q_{\phi_1}(z_{1:K}|\mathcal{D}_\tau^C)$;

6      **for** *state $s$ of each time step* **do**

7          Sample $e \sim q_{\phi_2}(e|s, z_{1:K})$;

8          Gather data from $\pi_\varphi(a|[s, z_{1:K}, e])$ to update $\mathcal{M}_\tau^C$;

9          Update $\mathcal{D}_\tau^C = \{(s_j, a_j, s_j', r_j)\}_{j=1}^N \sim \mathcal{M}_\tau^C$;

10      **end**

11 **end**

# B   Frequently Asked Questions

Here we collect some frequently asked questions from reviewers and other literature researchers. We thank these reviewers for these precious questions and add more explanation.

**Selection of Benchmarks.** Admittedly, NP variants can be applied a series of downstream tasks. Our selection of benchmark missions is based on existing literature for NP models. The system identification task was previously investigated with NP variants in work [43; 47], in which learning transferable physics dynamics with NPs is a crucial application. The image completion task is more commonly used in work [9; 2; 1]. The meta reinforcement learning task can also be studied within NP framework [3; 48].

**NP Family in Classification Tasks.** We have tried to search neural architectures for few-shot image classification tasks, but the performance is not ideal in comparison to other metrics based methods. Meanwhile, we have gone through most NP related work, and it is challenging to achieve SOTA few-shot image classification results with standard neural architectures in NPs, *e.g.* multi-layer perceptrons (MLPs). Maybe this is due to the nature of stochastic processes, which can address regression problems more efficiently. Unless specialize modules instead of MLPs are used, we do not expect NP variants with MLPs can achieve SOTA performance. The aim of this paper focuses more on mixture expert inductive biases and place less attention on neural architecture search. So MLPs are shared across all baselines to enable fair comparison.

**Expressiveness of Mixture of Expert Inductive Biases.** A natural question about learning diverse functional representation is whether these multiple expert latent variables will collapse into one. We refer the collapsed representation to vanilla (C)NPs, and the previous empirical results show the collapsed ones work poorer than mixture of expert ones in both few-shot regression tasks and meta reinforcement learning tasks. Also, from the multimodal simulation result, we discover both latent assignment variables and expert variables are meaningful, which reflects the effectiveness of mixture expert inductive biases. That means, we have not encountered meta representation collapse issues in experiments.

**Extension with other Mixture of Experts Models.** Our work is the first time to examine MoEs inductive biases in NPs family, and the used MoE module is an amortized inference one. We have not found a trivial implementation of MoEs in meta learning domain. But in MoEs literature, there exist other more effective MoE models, which can better trade off communication/memory and performance. So NPs family can also be combined with these models, such as GShard [49], Deepsepeed MoEs [50] and etc.

**Potential Applications in Industry.** Here we provide two available applications with MoE-NPs in the industry. One is in multilingual machine translation or multilingual language auto-completion. In this case, a mixture of experts corresponds to multilingual functional priors for multi-modal signals [51] and enables the prediction with partial observations. Another application lies in modelling irregular time series [52; 53]. In this case, diverse experts can handle discontinuous components in a rich family of stochastic functions. Meanwhile, the entropy of learned assignment latent variables can tell us the regions likely to be discontinuous, which is quite helpful in anomaly detection in a black-box system.

# C   Probabilistic Graphs in Meta Training/Testing

As exhibited in Fig. (10)/(11), shown are computational diagrams when implementing MoE-NPs in meta learning tasks.

**Variational Posteriors.** Since the real posteriors for both $\{z_k\}_{k=1}^K$ and $e_T$ are computationally intractable, the approximate ones are used in practice. These are called variational posteriors, e.g. $q_{\phi_{1,k}}$ for an expert latent variable $z_k$ and $q_{\phi_{2,1}}(e|x, y, z_{1:K})$ for assignment latent variables $e$. For the sake of simplicity, we denote $\{q_{\phi_{1,k}}\}_{k=1}^K$ by $q_{\phi_1}$ in some time. Importantly, the Gumbel-softmax trick [54] is used to sample assignment latent variable $e$ from categorical approximate distributions.

**Variational Priors.** In some cases, the prior distributions for $\{z_k\}_{k=1}^K$ and $e_T$ are set to constrain the scope of prior distributions. For example, in few-shot supervised learning, since the context and the target have the same form, the variational prior is selected to be $q_{\phi_1}$ as well to ensure the consistency

and this works in meta testing phases. For the assignment latent variable $e$, this uses the same form in conditional VAE [55] as $p_{\phi_{2,2}}$.

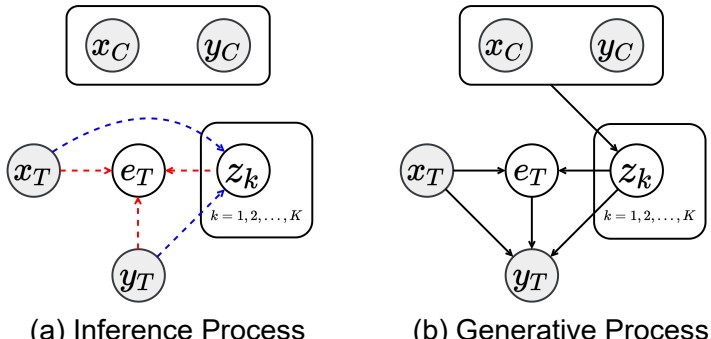

(a) Inference Process          (b) Generative Process

Figure 10: Computational Diagram in Few-shot Supervised Learning. Blue dotted lines are for expert latent variables while red dotted lines are for assignment latent variables in inference.

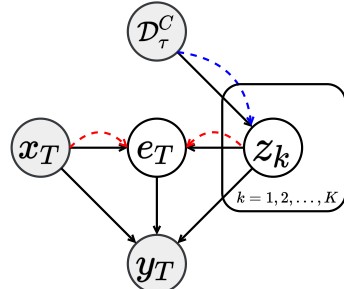

Figure 11: Computational Diagram in Meta Reinforcement Learning. This is in an information bottleneck form [6; 3]. The variational posteriors are $p_{\phi,k} = q_{\phi_{1,k}}(z|\mathcal{D}_\tau^C)$ and $q_{\phi_2} = q_{\phi_{1,k}}(e|s, z_{1:K})$. As for the variational prior, we use the same strategy as that in [6]. They are respectively the fixed normal $p(z) = \mathcal{N}(0, I)$ and the categorical $p(e) = \text{Cat}(e; K, [\frac{1}{K}, \frac{1}{K}, \ldots, \frac{1}{K}])$.

# D   More Descriptions of NP Family Models and Meta RL

In the main paper, we unify the description of NP family models in both few-shot supervised learning and meta reinforcement learning. This is the same with that in FCRL [3]. Meta learning in NP related models is to learn functional representations of different tasks and formulate the fast adaptation via inferring the task specific conditional distribution $p(\mathcal{D}_\tau^T|\mathcal{D}_\tau^C) = \int p(\mathcal{D}_\tau^T|z)p(z|\mathcal{D}_\tau^C)dz$ (equivalent to Eq. (1)).

To make the downstream reinforcement learning task using NP family models clearer, we add the following explanations. In few-shot supervised learning, $\mathcal{D}_\tau^C$ and $\mathcal{D}_\tau^T$ are of the same form. However, in context-based meta reinforcement learning, $\mathcal{D}_\tau^C$ is a set of task specific transitions and $\mathcal{D}_\tau^T$ is a set of state (action) values. As the result, the approximate posteriors and the selected priors to resolve Eq. (1) are distinguished in separate meta learning cases.

In context-based meta reinforcement learning, we can translate our problem into finding the distribution of optimal value functions in Eq. (5), this corresponds to learning meta critic modules with NP family models. Given a transition sample $[s, a, r(s, a), s']$, the target input is the state $x_T = s$ and the target output is the temporal difference target $y_T = \hat{Q}(s, a) = r(s, a) + \gamma V([s', z'])$. The standard Gaussian distribution is used as the prior $p(z_k) = \mathcal{N}(0, I)$ in Eq. (14), while the approximate posterior is learned from $\mathcal{D}_\tau^C$ with permutation invariant functions. In total, sampling from the state dependent approximate posterior $z \sim q_\phi(z|s, \mathcal{D}_\tau^C)$ corresponds to Eq. (15), where the operator $\odot$ denotes the selection process with help of Hadamard products.

$$z_{1:K} \sim q_{\phi_1}(z_{1:K}|\mathcal{D}_\tau^C), \ e \sim q_{\phi_2}(e|s, z_{1:K}), \ z = z_{1:K} \odot e \qquad (15)$$

# E MoE-NPs as Exchangeable $\mathcal{SP}$s

## E.1 Generative Processes

To better understand our developed model in meta learning scenarios, we translate Eq. (6) into a step-wise generative process. The same with that in the main paper, the task distribution is denoted by $p(\mathcal{T})$ and we presume $K$-experts to summarize the stochastic traits of a task.

$$\tau \sim p(\mathcal{T}), \quad z_k \sim p(z_k|\mathcal{T}) \; \forall k \in \{1, 2, \ldots, K\} \tag{16a}$$

$$x \sim p(x), \quad e \sim \prod_{k=1}^{K} \alpha_k(x, z_{1:K})^{\mathbb{I}[e_k=1]}, \quad z = [z_1, z_2, \ldots, z_K]^T \odot e \tag{16b}$$

$$[\mu_x, \Sigma_x] = g_\theta(x, z), \quad y \sim \mathcal{N}(\mu_x, \Sigma_x + \epsilon^2 I) \tag{16c}$$

Here a MoE-NP for the task $\tau$ is specified with $K$-expert latent variables $z_{1:K}$ in Eq. (16.a). The probability mass function for a data point related categorical distribution $\texttt{Cat}(K, \alpha(x, z_{1:K}))$ is denoted by $p(e|x, z_{1:K}) = \prod_{k=1}^{K} \alpha_k(x, z_{1:K})^{\mathbb{I}[e_k=1]}$ in Eq. (16.b), and $e$ is an assignment latent variable to select an expert for the generative process. After that, the distributional parameters for the output of a data point are learned via a function $g_\theta$ in Eq. (16.c), followed by the output distribution $\mathcal{N}(\mu_x, \Sigma_x + \epsilon^2 I)$.

Note that a collection of sampled functional experts are represented in a vector of variables $z_{1:K} = [z_1, z_2, \ldots, z_K]^T$ and $e = [0, \cdots, \underbrace{1}_{k\text{-th position}}, \cdots, 0]^T \Leftrightarrow e_k = 1$ is a one-hot vector in Eq. (16.b). In Eq. (16.c), the expert is selected in a way $z = z_{1:K} \odot e$. For the sake of generality, irreducible noise $\mathcal{N}(0, \epsilon^2 I)$ is injected in the output. In experiments, $K$-expert latent variables $z_{1:K}$ as well as discrete assignment latent variables $e$ are non-observable.

## E.2 Consistency Properties

**Definition 1. (Exchangeable Stochastic Process)** Given a probability space be $(\Omega, \mathcal{F}, \mathbb{P})$, let $\mu_{x_1,\ldots,x_N}$ be a probability measure on $\mathbb{R}^d$ with $\{x_1, \ldots, x_N\}$ as a finite index set. The defined process is called an exchangeable stochastic process ($\mathcal{SP}$), $\mathcal{S} : X \times \Omega \to \mathbb{R}^d$ such that $\mu_{x_1,\ldots,x_N}(F_1 \times \cdots \times F_N) = \mathbb{P}(\mathcal{S}_{x_1} \in F_1, \ldots, \mathcal{S}_{x_N} \in F_N)$ when it satisfies the exchangeable consistency and marginalization consistency.

Remember that the generative model is induced in the main paper as follows. And we claim that *our designed generative model MoE-NP formulates a family of exchangeable $\mathcal{SP}$ in* **Definition 1**.

$$\rho_{x_{1:N}}(y_{1:N}) = \int \prod_{k=1}^{K} p(z_k) \prod_{i=1}^{N} \left[ \sum_{k=1}^{K} p(y_i|x_i, z_{1:K}, e_k = 1)p(e_k = 1|x_i, z_{1:K}) \right] dz_{1:K} \tag{17}$$

So it is necessary to verify two formerly mentioned consistencies according to Kolmogorov Extension Theorem [35] and de Finetti's Theorem [56]. This is to show the existence of $\mathcal{SP}$s in Eq. (17).

**Exchangeability Consistency.** For $N$ data points from Eq. (17), we impose any permutation operation $\sigma$ over their indices, and this results in $\sigma : [1, 2, \ldots, N] \to [\sigma_1, \sigma_2, \ldots, \sigma_N]$. Then we can check the following equation is satisfied since the element-wise product of probabilities can be swapped.

$$\rho_{x_{1:N}}(y_{1:N}) = \int \left[ \prod_{k=1}^{K} p(z_k) \right] dz_{1:K} \prod_{i=1}^{N} \left[ \sum_{k=1}^{K} p(y_{\sigma_i}|x_{\sigma_i}, z_{1:K}, e_k = 1)p(e_k = 1|x_{\sigma_i}, z_{1:K}) \right]$$

$$= \int \left[ \prod_{k=1}^{K} p(z_k) \right] \prod_{i=1}^{N} \left[ \sum_{k=1}^{K} p(y_{\sigma_i}|x_{\sigma_i}, z_{1:K}, e_k = 1)p(e_k = 1|x_{\sigma_i}, z_{1:K}) \right] dz_{1:K}$$

$$= \rho_{x_{\sigma(1:N)}}(y_{\sigma(1:N)}) \quad \square$$

$$\tag{18}$$

**Marginalization Consistency.** Given the assumption that the integral in Eq. (17) is finite, we pick up a subset of indices $[M + 1, M + 2, \ldots, N]$ and make $M < N$ without difference in orders. And the result after marginalization over $y$-variable in the selected indices can be verified based on the following equation.

$$\int \rho_{x_{1:N}}(y_{1:N})dy_{M+1:N} = \int \left[ \prod_{k=1}^{K} p(z_k) \right] dz_{1:K} \left[ \int \prod_{i=1}^{N} p(y_i|x_i, z_{1:K})dy_{M+1:N} \right]$$

$$= \iint \prod_{i=1}^{M} p(y_i|x_i, z_{1:K}) \prod_{i=M+1}^{N} (p(y_i|x_i, z_{1:K}) \prod_{k=1}^{K} p(z_k)dz_{1:K}dy_{M+1:N}$$

$$= \int \left[ \prod_{k=1}^{K} p(z_k) \right] dz_{1:K} \prod_{i=1}^{M} \left[ \sum_{k=1}^{K} p(y_i|x_i, z_{1:K}, e_k = 1)p(e_k = 1|x_i, z_{1:K}) \right] = \rho_{x_{1:M}}(y_{1:M}) \quad \square$$

$$\tag{19}$$

Built on these two sufficient conditions, our developed MoE-NP is a well defined exchangeable $\mathcal{SP}$.

## F   Summary of Existing NP Related Models

### F.1   Comparison in Technical Details

Here we give a brief summary on difference between MoE-NPs and existing typical Neural Process models in Table (2). Some crucial traits include forms of encoders and decoders structures, types of latent variable and inductive biases injected in modelling. Especially, the inductive bias for MoE-NP is reduced to be multiple functional priors, which means a collection of expert neural processes to induce the generated dataset. Since a general inductive bias behind NPs related models is the modelling of exchangeable stochastic processes with cheap computations. This corresponds to a distribution of functions, termed as `functional` in the Table. Note that the recognition model of NP models in Meta Training Scenarios is replaced with $q_\phi(z|[x_T, y_T])$ since all target points can be available, but in Meta Testing Scenarios, only $[x_C, y_C]$ are accessible.

### F.2   Time Complexity

As for running time complexity, the vanilla NPs and CNPs are with $\mathcal{O}(N + M)$, while MoE-NPs are with $\mathcal{O}(K * (N + M))$ (making $M$ predictions with $N$ observations). In practice, the number of experts is small, so the increase of running time complexity can be ignored in practice. In contrast, traditional Gaussian processes are $\mathcal{O}((N + M)^3)$ in terms of running time complexity.

### F.3   Additional Literature Review

Due to page limit in the main paper, we include other related works in this subsection. In unsupervised learning, the Neural Statistician Model [57] is introduced to compute summary statistics inside the dataset. The Generative Query Network [48], a variant of NPs for visual sensory dataset, makes use of a latent variable to abstract scenes in high dimensions. To capture heteroscedastic noise inside the stochastic process, DSVNP [58] induces latent variables at different levels. The functional neural

Table 2: Summary of Typical Neural Process Related Models (Meta-Testing Scenarios). The recognition model and the generative model respectively correspond to the encoder and the decoder in the family of neural processes.

| Models | Recognition Model | Generative Model | Latent Variable | Inductive Bias |
|--------|-------------------|------------------|-----------------|----------------|
| CNP [2] | $z = f_\phi(x_C, y_C)$ | $p_\theta(y|[x, z])$ | continuous | conditional functional |
| NP [1] | $q_\phi(z|[x_C, y_C])$ | $p_\theta(y|[x, z])$ | continuous | global functional |
| ANP [14; 15] | $q_{\phi_1}(z|[x_C, y_C])$ $f_{\phi_*}(z_*|[x_C, y_C], x_*)$ | $p_\theta(y|[x, z, z_*])$ | continuous continuous | global functional local embedding |
| FCRL [3] | $f_\phi(z|[x_C, y_C])$ | $p_\theta(y|[x, z])$ | continuous | contrastive functional |
| ConvNP [16] | $p_\phi(z|[x_C, y_C])$ | $p_\theta(y|[x, z])$ | continuous | convolutional functional |
| Conv-CNP [17] | $f_\phi(z_*|[x_C, y_C], x_*)$ | $p_\theta(y|[x, z_*])$ | continuous | convolutional functional |
| MoE-NP (Ours) | $q_{\phi_1}(z_{1:K}|[x_C, y_C])$ $q_{\phi_{2,1}}(e|z_{1:K}, x, y)$ | $p_\theta(y|[x, z_{1:K}, e])$ $p_{\phi_{2,2}}(e|z_{1:K}, x)$ | continuous categorical | multiple functional |

processes infer the directed acyclic graph in the latent space and formulate flexible exchangeable stochastic processes for single task problems [59]. Inspired by self-supervised learning, [3; 60] propose to augment the neural process with contrastive losses. [61] combines context memories and recurrent memories to formulate sequential neural processes (SNPs). Though there exist a number of NP variants, none of them consider to inject multiple functional inductive bias in modeling.

## G   Formulation of Evidence Lower Bounds

Since functional priors reflected in the $K$-expert latent variables $z_{1:K}$ are learned via approximate distributions, this can be directly optimized within the framework of variational inference. So we leave these out in this discussion. The difficulty of optimization principally comes from the involvement of discrete latent variables. We therefore discuss chance of using another traditional optimization algorithm, called Expectation Maximization (EM) [62], in our settings. Omitting the $K$-expert latent variables $z_{1:K}$ and corresponding variational distributions, we take a closer look at the assignment latent variable $e$ in the logarithm likelihood as $\ln\left(\sum_{k=1}^{K} p(y|x, z_{1:K}, e_k = 1)p(e_k = 1|x, z_{1:K})\right)$ and derive the corresponding EM algorithm.

**Expectation(E)-Step**: Note that the assignment variable $e$ is discrete with the categorical probability function $p(e|x, z_{1:K}) = \texttt{Cat}(e; K, \alpha(x, z_{1:K}))$. This step is to update the posterior of the proportional coefficients $\alpha(x, z_{1:K})$ based on the last time step model parameters $\theta^{(t)}$.

$$\alpha_k^{(t+1)} = p(e_k = 1|x, z_{1:K}, y) \propto p(e_k = 1)p_{\theta^{(t)}}(y|x, z_{1:K}, e_k = 1) \tag{20}$$

Here the prior distribution $p(e)$ can be a commonly used one $\texttt{Cat}(K, [\frac{1}{K}, \frac{1}{K}, \ldots, \frac{1}{K}])$ or the last time updated one $p^{(t)}(e)$. As a result, updated categorical distribution parameters are:

$$\alpha^{(t+1)} = \begin{bmatrix} \alpha_1^{(t+1)} \\ \alpha_2^{(t+1)} \\ \vdots \\ \alpha_K^{(t+1)} \end{bmatrix} = \begin{bmatrix} \frac{\exp((\ln(p_{\theta^{(t)}}(y|x,z_{1:K},e_1=1)))/\tau)}{\sum_{k=1}^{K} \exp((\ln(p_{\theta^{(t)}}(y|x,z_{1:K},e_k=1)))/\tau)} \\ \frac{\exp((\ln(p_{\theta^{(t)}}(y|x,z_{1:K},e_2=1)))/\tau)}{\sum_{k=1}^{K} \exp((\ln(p_{\theta^{(t)}}(y|x,z_{1:K},e_k=1)))/\tau)} \\ \vdots \\ \frac{\exp((\ln(p_{\theta^{(t)}}(y|x,z_{1:K},e_K=1)))/\tau)}{\sum_{k=1}^{K} \exp((\ln(p_{\theta^{(t)}}(y|x,z_{1:K},e_k=1)))/\tau)} \end{bmatrix} \tag{21}$$

where $\tau$ is the temperature parameter.

**Maximization(M)-Step**:   Once the distributional parameter of assignment latent variables are updated, the next step is to maximize the logarithm likelihood as $\theta^{(t+1)} =$

$\arg\max_\theta \sum_{(x,y)\in\mathcal{D}} \ln\left[p_{\theta^{(t)}}(y|x,z_{1:K},e)\right]$ given the last time updated model parameter $\theta^{(t)}$. With help of gradient ascent, this can be written as follows,

$$\theta^{(t+1)} \leftarrow \theta^{(t)} + \lambda \sum_{(x,y)\in\mathcal{D}} \nabla_\theta \ln\left[p_{\theta^{(t)}}(y|x,z_{1:K},e)\right], \ e = \texttt{one\_hot}[\arg\max_k \alpha^{(t+1)}] \ \forall(x,y) \in \mathcal{D} \tag{22}$$

where $\lambda$ is the learning rate.

Note that the coefficient $\alpha$ is data point dependent and the derivation of EM algorithms considers a subset of data points $\mathcal{D}$. However, in meta learning scenarios, we handle large-scale dataset and the above-mentioned EM framework is computationally expensive and impractical. Due to these considerations, *we do not apply EM algorithms to estimate the discrete distribution* and instead variational inference is employed for the assignment latent variable in optimization.

### G.1 Variational Distributions

For continuous latent variables, diagonal Gaussians are commonly used as variational distributions. With Gaussian variational posteriors $\mathcal{N}(z;\mu,\Sigma)$ and corresponding priors $\mathcal{N}(z;\mu_p,\Sigma_p)$, the Kullback–Leibler Divergence can be analytically computed as follows.

$$D_{KL}[\mathcal{N}(z;\mu,\Sigma) \| \mathcal{N}(z;\mu_p,\Sigma_p)] = \frac{1}{2}[\ln\frac{|\Sigma_p|}{|\Sigma|} - d + (\mu-\mu_p)^T\Sigma_p^{-1}(\mu-\mu_p) + \texttt{Tr}\{\Sigma_p^{-1}\Sigma\}] \tag{23}$$

Meanwhile, when it comes to categorical distributions, the corresponding prior distribution is selected as $\texttt{Cat}(K,\alpha_0)$ with distribution parameters $\alpha_0 = [\alpha_{0,1},\alpha_{0,2},\ldots,\alpha_{0,K}]$. And the Kullback–Leibler Divergence is computed as follows.

$$D_{KL}[\texttt{Cat}(K,\alpha_*) \| \texttt{Cat}(K,\alpha_0)] = \sum_{k=1}^{K} \alpha_{*,k} \ln\left[\frac{\alpha_{*,k}}{\alpha_{0,k}}\right] \tag{24}$$

When $\alpha_0 = [\frac{1}{K}, \frac{1}{K}, \ldots, \frac{1}{K}]$, the divergence is further simplified as follows.

$$D_{KL}[\texttt{Cat}(K,\alpha_*) \| \texttt{Cat}(K,\alpha_0)] = \sum_{k=1}^{K} \alpha_{*,k} \ln\left[\frac{\alpha_{*,k}}{1/K}\right] = \sum_{k=1}^{K} \ln\alpha_{*,k} + \ln K \tag{25}$$

### G.2 Lower Bound on the Evidence for Few-Shot Supervised Learning

Since $K$-expert latent variables are independent in settings, we denote the corresponding variational parameters by $q_{\phi_1} = \{q_{\phi_{1,1}}, q_{\phi_{1,2}}, \ldots, q_{\phi_{1,K}}\}$. $\phi_{1,k}$ denotes parameters of encoders for $k$-th expert model. Hence, the distribution follows that $q_{\phi_1}(z_{1:K}|\mathcal{D}_\tau^C) = \prod_{k=1}^{K} q_{\phi_{1,k}}(z_k|\mathcal{D}_\tau^C)$ and $q_{\phi_1}(z_{1:K}|\mathcal{D}_\tau^T) = \prod_{k=1}^{K} q_{\phi_{1,k}}(z_k|\mathcal{D}_\tau^T)$.

Note that $(x,y) \in \mathcal{D}_\tau^T$ and the variational posterior for expert latent variables are $q_{\phi_1}(z_{1:K}|\mathcal{D}_\tau^T)$ in the general NPs. As for the variational posterior for assignment latent variables, we choose $q_{\phi_{2,1}}(e|x,y,z_{1:K})$ as default. We will use these notations to formulate the evidence lower bound (ELBO) as follows.

$$\ln p(y|x, \mathcal{D}_\tau^C) = \ln \int p(y|x, z_{1:K})p(z_{1:K}|\mathcal{D}_\tau^C)dz_{1:K}$$

(26a)

$$\geq \mathbb{E}_{q_{\phi_1}(z_{1:K}|\mathcal{D}_\tau^T)}\left[\ln p(y|x, z_{1:K})\right] - D_{KL}[q_{\phi_1}(z_{1:K}|\mathcal{D}_\tau^T) \parallel p(z_{1:K}|\mathcal{D}_\tau^C)]$$

(26b)

$$= \mathbb{E}_{q_{\phi_1}(z_{1:K}|\mathcal{D}_\tau^T)}\left[\ln \sum_{k=1}^{K} p(y, e_k = 1|x, z_{1:K})\right] - D_{KL}[q_{\phi_1}(z_{1:K}|\mathcal{D}_\tau^T) \parallel p(z_{1:K}|\mathcal{D}_\tau^C)]$$

(26c)

$$= \mathbb{E}_{q_{\phi_1}(z_{1:K}|\mathcal{D}_\tau^T)}\left[\ln \sum_{k=1}^{K} p(y|x, z_k)p(e_k = 1|x, z_{1:K})\right] - D_{KL}[q_{\phi_1}(z_{1:K}|\mathcal{D}_\tau^T) \parallel p(z_{1:K}|\mathcal{D}_\tau^C)]$$

(26d)

$$\geq \mathbb{E}_{q_{\phi_1}(z_{1:K}|\mathcal{D}_\tau^T)}\left[\mathbb{E}_{q_{\phi_{2,1}}(e|x,y,z_{1:K})}\left[\ln p_\theta(y|x, z_{1:K}, e)\right]\right]$$

(26e)

$$-\mathbb{E}_{q_{\phi_1}(z_{1:K}|\mathcal{D}_\tau^T)}\left[D_{KL}[\underbrace{q_{\phi_{2,1}}(e|x, y, z_{1:K})}_{\text{variational discrete posteriors}} \parallel p(e|x, z_{1:K})]\right]$$

(26f)

$$-\sum_{k=1}^{K} D_{KL}[\underbrace{q_{\phi_{1,k}}(z_k|\mathcal{D}_\tau^T)}_{K \text{ functional experts}} \parallel p(z_k|\mathcal{D}_\tau^C)]$$

(26g)

$$\approx \mathbb{E}_{q_{\phi_1}(z_{1:K}|\mathcal{D}_\tau^T)}\left[\mathbb{E}_{q_{\phi_{2,1}}(e|x,y,z_{1:K})}\left[\ln p_\theta(y|x, z_{1:K}, e)\right]\right]$$

(26h)

$$-\mathbb{E}_{q_{\phi_1}(z_{1:K}|\mathcal{D}_\tau^T)}\left[D_{KL}[\underbrace{q_{\phi_{2,1}}(e|x, y, z_{1:K})}_{\text{variational discrete posteriors}} \parallel \underbrace{p_{\phi_{2,2}}(e|x, z_{1:K})}_{\text{variational discrete priors}}]\right]$$

(26i)

$$-\sum_{k=1}^{K} D_{KL}[\underbrace{q_{\phi_{1,k}}(z_k|\mathcal{D}_\tau^T)}_{K \text{ functional experts}} \parallel q_{\phi_{1,k}}(z_k|\mathcal{D}_\tau^C)] = -\mathcal{L}(\theta, \phi_1, \phi_2) \quad \square$$

(26j)

By introducing the variational distribution $q_{\phi_2}$ for the discrete assignment latent variable $e$, Eq.(26.d) is further bounded by Eq. (26.e-g). Recall that when vanilla NP modules are used here, the approximate posterior in Eq. (26) in meta training should be substituted with $q_{\phi_1}(z_{1:K}|\mathcal{D}_\tau^T)$ with the corresponding approximate prior $p(z_k) = q_{\phi_{1,k}}(z_k|\mathcal{D}_\tau^C)$. And this matches the general form in the main paper for $-\mathcal{L}(\theta, \phi_1, \phi_2)$ in Eq. (7). When Dirac delta distributions are used in MoE-NPs, the divergence term about the continuous latent variable is removed as default. Denoting the approximate posterior by $q_{\phi_{2,1}}(e|x, y, z_{1:K}) = \mathtt{Cat}(e; [\alpha_1(x, y, z_{1:K}), \alpha_2(x, y, z_{1:K}), \ldots, \alpha_K(x, y, z_{1:K})])$, we rewrite the log-likelihood inside the ELBO as Eq. (27).

$$\mathbb{E}_{q_{\phi_{2,1}}(e|x,y,z_{1:K})}\left[\ln p(y|x, z_{1:K}, e)\right] = \sum_{k=1}^{K} \alpha_k \ln p(y|x, z_k)$$

(27)

As for the approximate posterior of the assignment latent variable $q_{\phi_{2,1}}(e|x, y, z_{1:K})$, we provide two ways of implementations in our experiments: (i) use the target input $y$ as the additional input to formulate $q_{\phi_{2,1}}(e|x, y, z_{1:K})$ (ii) use the same form as the conditional prior $q_{\phi_{2,1}}(e|x, y, z_{1:K}) = p_{\phi_{2,2}}(e|x, z_{1:K})$.

### G.3   Selection of Categorical Approximate Posteriors/Priors

As previously observed in Acrobot system identification results, increasing the number of expert latent variables tends to weaken the generalization capability. This also happens in image completion, so we set the number of experts used is 2 in the task. We can attribute this to inference sub-optimality in categorical approximate posteriors/priors.

Remember that in image completion and Acrobot system identification, the used approximate posterior for the categorical latent variable is $q_{\phi_{2,1}}(e|x, y, z_{1:K})$ with the target information $y$ for the input. Since the developed MoE-NP is a VAE-like models [30], the number of expert latent variables $K$ decides the dimension of the assignment latent variable $e$. In auto-encoder models, when the dimension of latent variables in all hidden layers is higher than that of the input, the model tends to copy the input to the output and fails to learn effective representations. This is the direct source of overfitting and applies to conditional VAE methods [55]. For example, the dimension of output in Acrobot is 6, which implies the bottleneck constraint is weaker when the number of experts is greater than 6.

It is reasonable to alleviate such sub-optimality by directly using the conditional prior as the approximate posterior $q_{\phi_{2,1}}(e|x, y, z_{1:K}) = p_{\phi_{2,2}}(e|x, z_{1:K})$. You can find more clues from the following stochastic gradient estimates for the assignment latent variables in Eq. (31). Meanwhile, we report empirical results in image completion when $q_{\phi_{2,1}}(e|x, y, z_{1:K}) = p_{\phi_{2,2}}(e|x, z_{1:K})$ in Sec. (I.1). And you can see inference in this way does not suffer the overfitting issue caused by more experts.

### G.4   Stochastic Gradient Estimates

Here the stochastic gradient estimates with respect to parameters in the negative ELBO $\mathcal{L}(\theta, \phi_1, \phi_2)$ in Eq. (7) are provided as follows.

$$\frac{\partial}{\partial \theta}\mathcal{L}(y; x, \theta, \phi_1, \phi_2) = \mathbb{E}_{q_{\phi_1}(z_{1:K}|\mathcal{D}_\tau^T)} \sum_{k=1}^{K} q_{\phi_{2,1}}(e_k = 1|x, y, z_{1:K}) \frac{\partial}{\partial \theta} \ln p_\theta(y|x, z_k) \tag{28}$$

$$\frac{\partial}{\partial \phi_{1,k}}\mathcal{L}(y; x, \theta, \phi_1, \phi_2) = \int \left[ \frac{\partial}{\partial \phi_{1,k}} q_{\phi_{1,k}}(z_k|\mathcal{D}_\tau^T) \right] \ln p_\theta(y|x, z_k) dz_k$$
$$- \frac{\partial}{\partial \phi_{1,k}} D_{KL}[q_{\phi_{1,k}}(z_k|\mathcal{D}_\tau^T) \parallel q_{\phi_{1,k}}(z_k|\mathcal{D}_\tau^C)] \tag{29}$$

$$\frac{\partial}{\partial \phi_2}\mathcal{L}(y; x, \theta, \phi_1, \phi_2) = \mathbb{E}_{q_{\phi_1}(z_{1:K}|\mathcal{D}_\tau^T)} \left[ \sum_{k=1}^{K} \left[ \frac{\partial}{\partial \phi_2} q_{\phi_{2,1}}(e_k = 1|x, y, z_{1:K}) \right] \ln p_\theta(y|x, z_k) \right]$$
$$- \mathbb{E}_{q_{\phi_1}(z_{1:K}|\mathcal{D}_\tau^T)} \left[ \frac{\partial}{\partial \phi_2} D_{KL}[q_{\phi_{2,1}}(e|x, y, z_{1:K}) \parallel p_{\phi_{2,2}}(e|x, z_{1:K})] \right] \tag{30}$$

The reparameterization trick [30] is used to sample values from variational distributions of expert latent variables throughout the inference process and stochastic gradient estimates in Eq. (28)/(29)/(30). In prediction processes, the way to get values of assignment latent variables follows that in [63].

Besides, we provide the stochastic gradient estimate for another case when the variational posterior for the assignment latent variable is selected as the variational prior, which means $q_{\phi_{2,1}}(e|x, y, z_{1:K}) = p_{\phi_{2,2}}(e|x, z_{1:K})$. This case can drop off the divergence term for the discrete variable. Let the conditional prior for the discrete variable be $p_{\phi_{2,2}}(e|x, z_{1:K}) =$

$\texttt{Cat}(e; [\alpha_1(x, z_{1:K}), \alpha_2(x, z_{1:K}), \dots, \alpha_K(x, z_{1:K})])$, we apply the log-derivative trick in a REIN-FORCE estimator [64] to Eq. (30) and can obtain the following equation as the gradient estimator[1].

$$
\begin{aligned}
\frac{\partial}{\partial \phi_{2,2}} \mathcal{L}(y; x, \theta, \phi_1, \phi_2) &= \mathbb{E}_{q_{\phi_1}(z_{1:K}|\mathcal{D}_\tau^T)} \left[ \sum_{k=1}^{K} \left[ \frac{\partial}{\partial \phi_{2,2}} p_{\phi_{2,2}}(e_k = 1|x, z_{1:K}) \right] \ln p_\theta(y|x, z_k) \right] \\
&= \mathbb{E}_{q_{\phi_1}(z_{1:K}|\mathcal{D}_\tau^T)} \left[ \sum_{k=1}^{K} p_{\phi_{2,2}}(e_k = 1|x, z_{1:K}) \left[ \underbrace{\frac{\partial}{\partial \phi_{2,2}} \ln p_{\phi_{2,2}}(e_k = 1|x, z_{1:K}) \ln p_\theta(y|x, z_k)}_{\text{Score Function}} \right] \right] \\
&= \mathbb{E}_{q_{\phi_1}(z_{1:K}|\mathcal{D}_\tau^T)} \left[ \sum_{k=1}^{K} \alpha_k \frac{\partial}{\partial \phi_{2,2}} \ln p_{\phi_{2,2}}(e_k = 1|x, z_{1:K}) \ln p_\theta(y|x, z_k) \right]
\end{aligned}
$$
(31)

As can be seen from Eq. (31), the posterior update implicitly exploits supervision information.

### G.5 Estimates of Statistics

**Momentum.** Given the pre-trained MoE-NPs, we can formulate the statistical momentum in predictive distributions. For the first order momentum, equivalently mean of the predictive distribution, we need to compute the conditional version $\mathbb{E}[Y|X = x, \mathcal{D}_\tau^C]$ in meta learning scenarios. Here the predictive distribution of one expert is parameterized in the form $p(y|x, z_k) = \mathcal{N}(y; m(x, z_k), \Sigma_k)$, where $m$ is the learned mean function using a neural network and $\sigma^2$ is a variance parameter. Using a single stochastic forward pass in expert latent variable $z_{1:K}$, we can derive the estimate of the predictive mean $\hat{m} = \mathbb{E}[Y|X = x, \mathcal{D}_\tau^C]$.

$$
\hat{m} = \sum_{k=1}^{K} \alpha_k \cdot m(x, z_k), \; \alpha_k = p_{\phi_{2,2}}(e_k = 1|z_{1:K}, x)
$$
(32)

The second order moment can be estimated accordingly. Here we consider the case when the output is one dimensional and $\Sigma_k = \sigma_k^2$.

$$
\mathbb{V}[Y|X = x, \mathcal{D}_\tau^C] = \mathbb{E}[Y^2] - \mathbb{E}[Y]^2 = \sum_{k=1}^{K} \alpha_k(\sigma_k^2 + m(x, z_k)^2) - \hat{m}^2
$$
(33)

**Entropy.** Note that our developed MoE-NPs can also be applied to out of detection (O.O.D) tasks. In this case, the entropy of predictive distribution plays a crucial role. Though the exact estimate of the predictive entropy for MoE-NPs is intractable due to the complexity inside the mixture components, we can measure the expected result of the entropy $\mathbb{E}[\mathcal{H}(Y)|X = x, \mathcal{D}_\tau^C]$ in prediction. We still use a single stochastic forward pass in expert latent variable $z_{1:K}$ in estimation. If $\mathcal{H}(Y_k|X = x, z_k) = -\int p(Y = y|X = x, z_k) \ln p(Y = y|X = x, z_k) dY$ is bounded $\forall k \in \{1, \dots, K\}$, the estimate of entropy term is as follows.

$$
\hat{\mathbb{E}}[\mathcal{H}(Y)|X = x, \mathcal{D}_\tau^C] = \sum_{k=1}^{K} \alpha_k \int p(Y = y|X = x, z_k) \mathcal{H}(Y = y) dy = \sum_{k=1}^{K} \alpha_k \mathbb{E}[\mathcal{H}(Y_k|X = x, z_k)]
$$
(34)

## H Experimental Settings and Neural Architectures

In this section, we provide with more experimental details. Importantly, neural modules of MoE-NPs in the PyTorch version are listed. For the few-shot regression, we provide an example of our implementation of MoE-NPs from the anonymous Github link

---

[1]For discrete latent variables, we can obtain the analytical form of the stochastic gradient.

## H.1   Dataset & Environments

### H.1.1   Dataset in Few-shot Supervised Learning

**System Identification.** Note that in the used Acrobot simulator [2], the observation is the pre-processed state as a 6 dimensional vector $[\sin(\theta_1), \cos(\theta_1), \sin(\theta_2), \cos(\theta_2), \theta_1', \theta_2']$. The input of the Acrobot system is the concatenation of the observation and the executed action $[\sin(\theta_1), \cos(\theta_1), \sin(\theta_2), \cos(\theta_2), \theta_1', \theta_2', a]$. The output of Acrobot system is the predicted transited state. We generate 16 meta training tasks by varying the masses of two pendulums $m_1$ and $m_2$, which means the hyper-parameters of the system come from the Cartesian combination of the set $m_1 \in \{0.75, 0.85, 0.95, 1.15\}$ and $m_2 \in \{0.75, 0.85, 0.95, 1.15\}$. In meta training processes, a complete random policy interacts with batch of sampled MDPs to formulate transition dataset. As for meta testing tasks, we follow the same way to generate tasks by setting $m_1 \in \{0.85, 1.05, 1.25\}$ and $m_2 \in \{0.85, 1.05, 1.25\}$.

**Image Completion.** CIFAR10 dataset consists of 60000 32x32 color images in 10 categories. Among these images, 50000 are for meta training with the rest for meta testing as the default in image completion tasks. CIFAR10 images are processed via torchvision modules to normalize the pixel values between $[0, 1]$.

### H.1.2   Environments in Meta Reinforcement Learning

Note that 2-D point robot navigation tasks, the distribution for meta training is a mixture of uniform distributions $[0, 2\pi/12] \cup [5\pi/12, 7\pi/12] \cup [10\pi/12, \pi]$. The rest of regions along the arc is for out of distribution tasks. The tasks in Mujoco [7] follows adaptations from [6; 65], where goals/velocities or multiple hyper parameters of simulation systems are sampled from mixture distributions. The horizon of an episode for mixture of point robots is 20, while that for Mujoco environments is 500. The required numbers of environment steps in meta training processes are respectively $2.5 * 1e6$ for point robots, $7.5 * 1e6$ for Half-Cheetah-CD and $6.5 * 1e6$ for Slim-Humanoid-CG. We leave more details and settings of each environment in the above github code link.

**2-D Point Robot.** 2-D robots attempt to reach goals located in specified regions of the arc.

**Cheetah-Complex-Direction.** 2-D Cheetah robots aim at running in given directions. The task includes multiple target directions and these change with split steps of episodes.

**Humanoid-Complex-Goals.** 3-D Humanoid robots aim at running towards goals. The task includes multiple goals and these change with split steps of episodes.

## H.2   Implementations in Meta Learning Tasks

### H.2.1   Toy Experiments

The input of functions is in a range $[-\pi, \pi] \cup [\pi, 2\pi]$. The general implementations of MoE-NPs are as follows. The $x$-domain for the function $f_1(x) = \sin(x) + \epsilon_1$ with $\epsilon_1 \sim \mathcal{N}(0, 0.03^2)$ is $[-\pi, \pi]$, while that for $f_2(x) = \cos(x) + \epsilon_2$ with $\epsilon_2 \sim \mathcal{N}(0, 0.01^2)$ is $[\pi, 2\pi]$. Sampling from these two components leads to a mixture dataset. The `Encoder` for all continuous latent variables is with two hidden layers (32 neuron units each). The Gaussian distribution is used for continuous latent variables in MoE-NPs. The `Encoder` for the discrete assignment latent variable in MoE-NPs corresponds to a softmax-output neural network with two hidden layers (16 neuron units each). The `Decoder` is with three hidden layers (128 neuron units each) as well. The number of data points in each sampled task is 100.

---

[2] `https://github.com/openai/gym/blob/master/gym/envs/classic_control/acrobot.py`

### H.2.2 Few-Shot Supervised Learning

For gradient based methods, we use implementations of MAML[3] and CAVIA[4].

**System Identification.** The general implementations are as follows. For all NP related models, the dimension of a latent variable is set to be 16. The `Encoder` for all continuous latent variables is with two hidden layers (32 neuron units each). Note that Dirac delta distributions are used for continuous latent variables in MoE-NPs since this case works best in few-shot supervised learning. For MoE-NPs, we use three expert latent variables as the default, and the `Encoder` for the discrete assignment latent variable in MoE-NPs corresponds to a softmax-output neural network with two hidden layers (32 neuron units each). The `Decoder` is with four hidden layers (200 neuron units each) as well. The number of tasks in batch training is 4, batch size in training is 200 transition steps (for each task). The horizon for each episode of transitions is 200 time steps. In each iteration of meta training, 4 different environments are randomly selected, and the uniform random controller is used to interact for the collection of 5 episodes (for each task). In training dynamics systems, the training batch size in transition buffer is 200, the training epoch is 5, and the whole process iterates until convergence (the iteration number is 25). The learning rate for Adam optimizer is $1e-3$ as the default.

**Image Completion.** The general implementations follow that in [9; 2] and are applied to all baselines and MoE-NPs. The dimension of a latent variable is set to be 128. The `Encoder` for all NP variants is with three hidden layers (128 neuron units each). Note that Dirac delta distributions are used for continuous latent variables in MoE-NPs since this case works best in few-shot supervised learning. The `Decoder` is with five hidden layers (128 neuron units each) as well. For MoE-NPs, we use three expert latent variables as the default, and the `Encoder` for the discrete assignment latent variable corresponds to a softmax-output neural network with two hidden layers (32 neuron units each). Adam [30] is used as the optimizer, where the learning rate is set to be $5e-4$. The batch size in training is 8 images, and we meta train the model until convergence (the maximum epoch number is 50). Also note that the number of context pixels in CAVIA is 10 for fast adaptation in default implementations, and this leads to the best testing result of CAVIA in 10 pixel cases in Fig. (4). Note that, to train NP models, including CNP/NP/FCRL/MoE-NP, we set the form of negative log-likelihood objective consistent based on that in [66]. But in evaluation, to keep results of all methods consistent, we follow that in [2; 9; 3] and report the MSEs in Fig. (4) in the testing phase.

### H.2.3 Meta Reinforcement Learning

In terms of implementations of baselines, we directly use the following open sourced code: PEARL[5], MAML[6] and CAVIA[7]. Note that the TRPO algorithm [67] is used for MAML/CAVIA as the default. We do not change too much except the replacement of our environments when running experiments.

Further we provide more details on how to modify MoE-NPs in meta RL domains. Notice that MoE-NP can also be seen as a latent variable model, and there exists a close relationship with PEARL algorithms [6] when it comes into meta RL. Implementations of MoE-NPs are the same as that in PEARL [6] except latent variable distributions and the inference way. Note that Soft Actor Critic (SAC) algorithm [39] is used in policy optimization, which requires parameterization of both actor and critic functions. As for the number of experts in MoE-NPs, we use 3 for all environments as default. You can find more details about neural architectures/optimizers for each modules from the above mentioned link.

As mentioned before, we use $p(z_k) = \mathcal{N}(0, I)$ as the prior distribution for expert latent variables. The approximate posterior is parameterized with a diagonal Gaussian distribution. The coefficient for KL divergence terms in Eq. (14) are $\beta_0 = 1.0, \beta_1 = 1.0$. The meta-training processes for reinforcement learning can be found in Algorithm (3) in the main paper. In terms of meta-testing processes, we report the pseudo code in Algorithm (4).

---

[3] https://github.com/cbfinn/maml_rl
[4] https://github.com/lmzintgraf/cavia
[5] https://github.com/katerakelly/oyster
[6] https://github.com/cbfinn/maml_rl
[7] https://github.com/lmzintgraf/cavia

## H.3 Neural Architectures

To help readers better understand our models, we attach the python code of the `Encoder` for separate latent variables as follows. As for the `Decoder`, the structure is the same with that in a vanilla NP/CNP [1; 2].

```python
import torch
import torch.nn as nn
import torch.nn.functional as F

#####################################################################
    # context encoders for expert latent variables
#####################################################################

class Context_Encoder(nn.Module):
    '''
    Encoder network for [x_c, y_c]
    '''
    def __init__(self,
                 input_size,
                 hidden_size,
                 act_type,
                 num_layers,
                 output_size):
        super(Context_Encoder, self).__init__()

        self.emb_c_modules = []
        self.emb_c_modules.append(nn.Linear(input_size,
                                            hidden_size))
        for i in range(num_layers):
            self.emb_c_modules.append(nn.ReLU())
            self.emb_c_modules.append(nn.Linear(hidden_size,
                                                hidden_size))
        self.emb_c_modules.append(nn.ReLU())
        self.context_net = nn.Sequential(*self.emb_c_modules)

        self.mu_net = nn.Linear(hidden_size,
                                output_size)
        self.logvar_net = nn.Linear(hidden_size,
                                    output_size)

    def forward(self, x, mean_dim=1):
        out = self.context_net(x)
        out = torch.mean(out,dim=mean_dim)

        mu, logvar = self.mu_net(out), self.logvar_net(out)

        return (mu,logvar)

#####################################################################
    # context encoders for discrete assignment latent variables
#####################################################################

class Softmax_Net(nn.Module):
    def __init__(self,
                 dim_xz,
                 experts_in_gates,
                 dim_logit_h,
                 num_logit_layers,
                 num_experts):
        super().__init__()
        self.dim_xz = dim_xz
```

```python
        self.experts_in_gates = experts_in_gates
        self.dim_logit_h = dim_logit_h
        self.num_logit_layers = num_logit_layers
        self.num_experts = num_experts

        self.logit_modules = []
        if self.experts_in_gates:
            self.logit_modules.append(nn.Linear(self.dim_xz,
                                                self.dim_logit_h))
            for i in range(self.num_logit_layers):
                self.logit_modules.append(nn.ReLU())
                self.logit_modules.append(nn.Linear(self.dim_logit_h,
                                                    self.dim_logit_h))
            self.logit_modules.append(nn.ReLU())
            self.logit_modules.append(nn.Linear(self.dim_logit_h,
                                                1))
        else:
            self.logit_modules.append(nn.Linear(self.dim_xz,
                                                self.dim_logit_h))
            for i in range(self.num_logit_layers):
                self.logit_modules.append(nn.ReLU())
                self.logit_modules.append(nn.Linear(self.dim_logit_h,
                                                    self.dim_logit_h))
            self.logit_modules.append(nn.ReLU())
            self.logit_modules.append(nn.Linear(self.dim_logit_h,
                                                self.num_experts))
        self.logit_net=nn.Sequential(*self.logit_modules)

    def forward(self, x_z, temperature, gumbel_max=False):

        if self.experts_in_gates:
            logit_output = self.logit_net(x_z)
        else:
            x_z = torch.mean(x_z, dim=-2)
            logit_output = self.logit_net(x_z)

        if not self.experts_in_gates:
            logit_output = logit_output.unsqueeze(-1)

        if gumbel_max:
            logit_output = logit_output \
                + sample_gumbel(logit_output.size())

        softmax_y = F.softmax(logit_output/temperature, dim=-2)

        softmax_y = softmax_y.squeeze(-1)
        shape = softmax_y.size()
        _,ind = softmax_y.max(dim=-1)
        y_hard = torch.zeros_like(softmax_y).view(-1, shape[-1])
        y_hard.scatter_(1, ind.view(-1, 1), 1)
        y_hard = y_hard.view(*shape)

        y_hard = (y_hard-softmax_y).detach() \
            + softmax_y

        softmax_y, y_hard = softmax_y.unsqueeze(-1), y_hard.unsqueeze(-1)

        return softmax_y, y_hard
```

Putting the mentioned structures together, we include the MoE-NPs as follows.

```python
##################################################################
    # mixture of expert neural processes
##################################################################

class MoE_NP(nn.Module):
    def __init__(self,args):
        super(MoE_NP,self).__init__()

        # extract parameters from args
        self.dim_x=args.dim_x
        self.dim_y=args.dim_y

        self.dim_h_lat=args.dim_h_lat
        self.num_h_lat=args.num_h_lat
        self.dim_lat=args.dim_lat
        self.num_lat=args.num_lat
        self.experts_in_gates=args.experts_in_gates
        self.num_logit_layers=args.num_logit_layers
        self.dim_logit_h=args.dim_logit_h
        self.temperature=args.temperature
        self.gumbel_max=args.gumbel_max
        self.info_bottleneck=args.info_bottleneck

        self.dim_h=args.dim_h
        self.num_h=args.num_h
        self.act_type=args.act_type
        self.amort_y=args.amort_y

        # encoding networks
        self.expert_modules=nn.ModuleList([Context_Encoder(self.dim_x+self.dim_y,
                                                    self.dim_h_lat,
                                                    self.act_type,
                                                    self.num_h_lat,
                                                    self.dim_lat).cuda()
                                      for i in range(self.num_lat)])

        if self.experts_in_gates:
            self.logit_net_post=Softmax_Net(self.dim_x+self.dim_y+self.dim_lat,
                                            self.experts_in_gates,
                                            self.dim_logit_h,
                                            self.num_logit_layers,
                                            self.num_lat)
            self.logit_net_prior=Softmax_Net(self.dim_x+self.dim_lat,
                                             self.experts_in_gates,
                                             self.dim_logit_h,
                                             self.num_logit_layers,
                                             self.num_lat)
        else:
            self.logit_net_post=Softmax_Net(self.dim_x+self.dim_y,
                                            self.experts_in_gates,
                                            self.dim_logit_h,
                                            self.num_logit_layers,
                                            self.num_lat)
            self.logit_net_prior=Softmax_Net(self.dim_x,
                                             self.experts_in_gates,
                                             self.dim_logit_h,
                                             self.num_logit_layers,
                                             self.num_lat)

        # decoding networks
        self.dec_modules=[]
        self.dec_modules.append(nn.Linear(self.dim_x+self.dim_lat,
                                    self.dim_h))
```

```python
        for i in range(self.num_h):
            self.dec_modules.append(get_act(self.act_type))
            self.dec_modules.append(nn.Linear(self.dim_h,
                                                self.dim_h))
        if self.amort_y:
            self.dec_modules.append(get_act(self.act_type))
            self.dec_modules.append(nn.Linear(self.dim_h,
                                                2*self.dim_y))
        else:
            self.dec_modules.append(get_act(self.act_type))
            self.dec_modules.append(nn.Linear(self.dim_h,
                                                self.dim_y))
        self.dec_net=nn.Sequential(*self.dec_modules)

    def get_context_idx(self,M):
        # generate the indeces of the N context points from M points
        N = random.randint(1,M)
        idx = random.sample(range(0, M), N)
        idx = torch.tensor(idx).cuda()

        return idx

    def idx_to_data(self,data,sample_dim,idx):
        # get subset of an array
        ind_data= torch.index_select(data, dim=sample_dim, index=idx)

        return ind_data

    def reparameterization(self,mu,logvar):
        std=torch.exp(0.5*logvar)
        eps=torch.randn_like(std)

        return mu+eps*std

    def encoder(self,x_c,y_c,x_t,y_t):
        if self.training:
            memo_c,memo_t=torch.cat((x_c,y_c),dim=-1),torch.cat((x_t,y_t),dim=-1)
            emb_c_list_mu=torch.cat([expert_module(memo_c)[0].unsqueeze(0)
                                    for expert_module
                                    in self.expert_modules])
            emb_c_list_logvar=torch.cat([expert_module(memo_c)[1].unsqueeze(0)
                                        for expert_module
                                        in self.expert_modules])
            emb_t_list_mu=torch.cat([expert_module(memo_t)[0].unsqueeze(0)
                                    for expert_module
                                    in self.expert_modules])
            emb_t_list_logvar=torch.cat([expert_module(memo_t)[1].unsqueeze(0)
                                        for expert_module
                                        in self.expert_modules])

            emb_c_list_mu,emb_c_list_logvar=emb_c_list_mu.permute(1,0,2),\
                emb_c_list_logvar.permute(1,0,2)
            emb_t_list_mu,emb_t_list_logvar=emb_t_list_mu.permute(1,0,2),\
                emb_t_list_logvar.permute(1,0,2)

        else:
            memo_c=torch.cat((x_c,y_c),dim=-1)
            emb_c_list_mu=torch.cat([expert_module(memo_c)[0].unsqueeze(0)
                                    for expert_module
                                    in self.expert_modules])
```

```python
            emb_c_list_logvar=torch.cat([expert_module(memo_c)[1].unsqueeze(0)
                                        for expert_module
                                        in self.expert_modules])
            emb_c_list_mu,emb_c_list_logvar=emb_c_list_mu.permute(1,0,2),\
                emb_c_list_logvar.permute(1,0,2)

            emb_t_list_mu,emb_t_list_logvar=0,0

        return emb_c_list_mu,emb_c_list_logvar,emb_t_list_mu,emb_t_list_logvar

    def forward(self,x_c,y_c,x_t,y_t,x_pred,y_pred,
                whether_acrobot=False,whether_image=True):
        mu_c,logvar_c,mu_t,logvar_t=self.encoder(x_c, y_c, x_t, y_t)

        if self.training:
            if self.info_bottleneck:
                z_experts=self.reparameterization(mu_t,logvar_t)
            else:
                z_experts=mu_c
        else:
            assert y_pred==None
            if self.info_bottleneck:
                z_experts=self.reparameterization(mu_c,logvar_c)
            else:
                z_experts=mu_c

        z_experts_unsq=z_experts.unsqueeze(1).expand(-1,x_pred.size()[1],-1,-1)

        x_exp=x_pred.unsqueeze(2).expand(-1,-1,z_experts_unsq.size()[2],-1)

        if self.training:
            y_exp=y_pred.unsqueeze(2).expand(-1,-1,z_experts_unsq.size()[2],-1)
            if self.experts_in_gates:
                xz_exp=torch.cat((x_exp,z_experts_unsq),dim=-1)
                xy_exp=torch.cat((x_exp,y_exp),dim=-1)
                xyz_exp=torch.cat((xy_exp,z_experts_unsq),dim=-1)
                alpha_post,y_hard_post=self.logit_net_post(x_z=xyz_exp,
                                                temperature=self.temperature,
                                                gumbel_max=self.gumbel_max)
                alpha_prior,y_hard_prior=self.logit_net_prior(x_z=xz_exp,
                                                temperature=self.temperature,
                                                gumbel_max=self.gumbel_max)
            else:
                xy_exp=torch.cat((x_exp,y_exp),dim=-1)
                alpha_post,y_hard_post=self.logit_net_post(x_z=xy_exp,
                                                temperature=self.temperature,
                                                gumbel_max=self.gumbel_max)
                alpha_prior,y_hard_prior=self.logit_net_prior(x_z=x_exp,
                                                temperature=self.temperature,
                                                gumbel_max=self.gumbel_max)
        else:
            if self.experts_in_gates:
                xz_exp=torch.cat((x_exp,z_experts_unsq),dim=-1)
                alpha_post,y_hard_post=0,0
                alpha_prior,y_hard_prior=self.logit_net_prior(x_z=xz_exp,
                                                temperature=self.temperature,
                                                gumbel_max=self.gumbel_max)
            else:
                alpha_post,y_hard_post=0,0
                alpha_prior,y_hard_prior=self.logit_net_prior(x_z=x_exp,
                                                temperature=self.temperature,
                                                gumbel_max=self.gumbel_max)
        output=self.dec_net(torch.cat((x_exp,z_experts_unsq),dim=-1))
```

```python
if whether_acrobot:
    if self.amort_y:
        y_mean,y_std=output[...,:self.dim_y],\
            F.softplus(output[...,self.dim_y:])
        return mu_c,logvar_c,mu_t,logvar_t,\
            y_mean,y_std,alpha_post,alpha_prior
    else:
        y_pred=torch.cat((torch.cos(output[...,0:1]),
                          torch.sin(output[...,0:1]),
                          torch.cos(output[...,1:2]),
                          torch.sin(output[...,1:2]),
                          4*pi*torch.tanh(output[...,2:3]),
                          9*pi*torch.tanh(output[...,3:4])),axis=-1)
        return mu_c,logvar_c,mu_t,logvar_t,\
            y_pred,alpha_post,alpha_prior
elif whether_image:
    if self.amort_y:
        y_mean,y_std=F.sigmoid(output[...,:self.dim_y]),\
            F.softplus(output[...,self.dim_y:])
        return mu_c,logvar_c,mu_t,logvar_t,\
            y_mean,y_std,alpha_post,alpha_prior
    else:
        y_pred=F.sigmoid(output)
        return mu_c,logvar_c,mu_t,logvar_t,\
            y_pred,alpha_post,alpha_prior
else:
    if self.amort_y:
        y_mean,y_std=output[...,:self.dim_y],\
            F.softplus(output[...,self.dim_y:])
        return mu_c,logvar_c,mu_t,logvar_t,\
            y_mean,y_std,alpha_post,alpha_prior
    else:
        y_pred=output
        return mu_c,logvar_c,mu_t,logvar_t,\
            y_pred,alpha_post,alpha_prior
```

# I Additional Experimental Results

## I.1 Additional Analysis of Learned Latent Variables

Here we give more detailed analysis *w.r.t.* learned latent variables in MoE-NPs.

**Entropy of Assignment Latent Variables.** Still we take the 1-dimensional toy stochastic function as example because it is intuitive to understand the latent variable meanings of different levels. Remember that the role of the discrete latent variable $e$ is to assign the diverse functional prior $z_{1:K}$ to a given data point. With the learned conditional prior $p_{\phi_{2,2}}(e|z_{1:K}, x_i)$ for a data point $x_i$, we can quantify the uncertainty of assignment via the entropy of such a Bernoulli latent variable $\mathcal{H}[e]$.

$$\mathcal{H}[e] = \sum_{k=1}^{K} -p_{\phi_{2,2}}(e_k = 1|z_{1:K}, x_i) \ln p_{\phi_{2,2}}(e_k = 1|z_{1:K}, x_i) = \sum_{k=1}^{K} -\alpha_k \ln \alpha_k \qquad (35)$$

This has a practical significance in discontinuous functions. For example, in regions close to demarcation points, it should be difficult to judge the best expert $z_{1:K}$ to handle these data points, which means the set of $\mathcal{H}[e]$ theoretically exhibits higher uncertainty. Similarly, in regions without context points, it is hard to determine the function as well.

Interestingly, we observe that MoE-NPs are able to exhibit the above effect on the right side of Fig. (12). The sampled function consists of two components respectively in the interval $[-\pi, \pi]$ and $[\pi, 3\pi]$. The entropy values of our interest are computed via Eq. (35). Here $K = 2$ and the learned conditional prior $p_{\phi_{2,2}}(e|z_{1:K}, x_i)$ has highest entropy around the demarcation data point $\pi$ and the

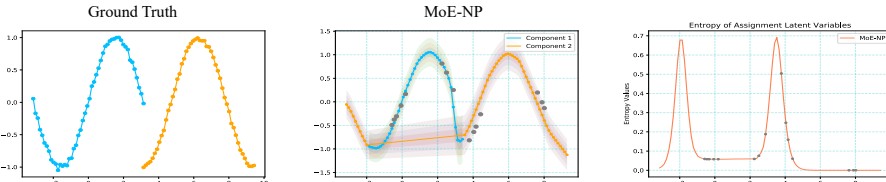

Figure 12: Entropy of Assignment Latent Variables in MoE-NPs. From left to right are respectively the sampled ground truth function, MoE-NP fitting results and the entropy value of discrete latent variable for each data point $\mathcal{H}[p_{\phi_{2,2}}(e|z_{1:K}, x_i)]$.

data point $-2.0$ with no context points nearby. This finding further verifies the role of the assignment latent variable in MoE-NPs.

**Number of Expert Latent Variables.** By setting the approximate posterior of assignment latent variables as $q_{\phi_{2,1}}(e|x, y, z_{1:K}) = p_{\phi_{2,2}}(e|x, z_{1:K})$, we further investigate the scalability issue of MoE-NPs in CIFAR10 image completion. As reported in Table (3), we can find with more experts ($>= 3$), the performance can be further improved and no overfitting issue occurs. It can also be inferred when the number of experts reaches a certain level, the improvement from the increase of expert numbers is quite limited. So in general, when the output dimension is lower, the best choice of the approximate posterior for assignment latent variables is in a form without $y$ as the input.

Table 3: Pixel-wise mean squared errors (MSEs) with varying number of experts in CIFAR10 image completion. The number of random context points is varied in a range $(10, 200, 500, 800, 1000)$ to test performance at different levels.

| # | 10 | 200 | 500 | 800 | 1000 |
|---|---|---|---|---|---|
| MoE-NPs (3 experts) | 0.0482 | 0.0183 | 0.0170 | 0.0166 | 0.0165 |
| MoE-NPs (5 experts) | 0.0362 | 0.0103 | 0.0071 | 0.0061 | 0.0057 |
| MoE-NPs (7 experts) | 0.0359 | 0.0095 | 0.0062 | 0.0052 | 0.0048 |

## I.2 Additional Results of NP Variants in Toy Regression

Note that the variation of tasks in the previous toy regression is quite limited and its goal is to show the role of latent variables. To further examine the performance, we construct the mixture of sinusoidal functions by varying the amplitude and the phase as follows.

The learning data points are randomly sampled in $x$-domain and merged from a mixture of randomized functions $f_1(x) = A\sin(x - B) + \epsilon$ in $x$-domain $[-\pi, \pi]$ and $f_2(x) = A\cos(x - B) + \epsilon$ in $x$-domain $[\pi, 3\pi]$ with equal probability, where $\epsilon \sim \mathcal{N}(0, 0.03^2)$. The range of the amplitude is $A \in [0.1, 5.0]$ while that of the phase is $B \in [0, \pi]$.

In each training iteration, we sample a batch of data points and randomly partition context points and target points for learning. Each task consists of 100 randomly sampled data points from the mixture of sinusoidal functions with the random number of context points between $[5, 50]$. The default number of tasks in a meta batch is 25 and we set the number of iteration steps at most 50000. As for neural architectures of all baselines, we retain that in the previous toy regression in Sec. (H.2.1). Still we use two experts for MoE-NPs as default to fit mixture sinusoidal functions.

Table 4: Test Performance in Mixture Sinusoidal Functions. Shown are mean square errors and standard deviations in fitting 500 sampled tasks. The best results in 5 runs are in bold with standard deviations in bracket.

| CNP | NP | FCRL | ANP | MoE-NP |
|---|---|---|---|---|
| 0.053($\pm$1E-4) | 0.070($\pm$3E-4) | 0.040($\pm$0.0) | **0.027($\pm$2E-4)** | 0.032(1E-4) |

In meta testing phase, we draw up 500 tasks with 15 random data points selected as the context. These testing tasks are generated in the way: the couple of the amplitude and the phase $[A, B]$ are orderly set from the amplitude list `numpy.linspace`$(0.1, 5.0, \text{num} = 500)$ and phase

list `numpy.linspace`$(0.0, \pi, \texttt{num} = 500)$. The sampled $x$-values for these tasks are a list `torch.linspace`$(-\pi, 3\pi, \texttt{steps} = 500)$. As shown in Table (4), ANP achieves best performance in meta testing, followed by MoE-NP.

### I.3 Additional Results of NP Variants in System Identification

To understand how performance evolves with more context transitions in Acrobot system, we extend the result of 50 context points in the main paper to Fig. (13). As can be seen, MoE-NP still outperforms all NP baselines in all cases. And with the increase of context transitions, we can find the predictive MSEs degrade accordingly.

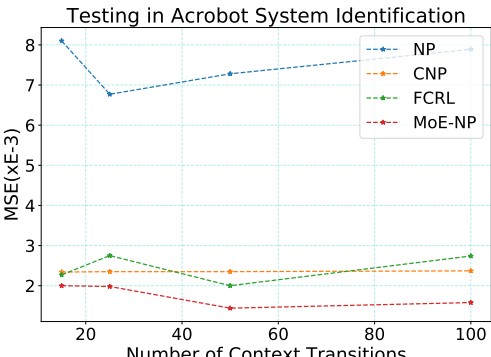

Figure 13: Asymptotic Performance in System Identification of NPs Family. The numbers of random transitions as the context are respectively $\{15, 25, 50, 100\}$.

### I.4 Comparison with Attentive Neural Processes

Since neural architectures for attentive neural processes (ANPs) [14] are bit different from used baselines and cannot be trivially modified to meta RL cases, we report additional results in this subsection.

We implement ANPs with dot attention networks (since ANPs have more model complexity and can easily lead to cuda out of memory in practice, we choose the basic version of ANPs), the input embedding dimension of to compute attention weights is 32 and 4 layers are used to transform the deterministic embedding $z_{\text{attn}}$. The local embedding is concatenated with the input $x$ and the global latent variable $z$ for the decoder. We use three heads for system identification tasks and one head for image completion tasks. The related results are reported as follows. It can be seen in Table (5), MoE-NPs still outperform ANPs, while ANPs beat NPs a lot in predicting Acrobot system dynamics. As for CIFAR10 image completion, we can draw the same conclusion in Table (6) that Mixture Expert inductive biases are more effective than local latent variables embedded in attention modules.

Table 5: System identification in Acrobot. Meta testing results are reported. We use the number of random transitions as the context and test performance for ANPs to compare. Figures in the Table are scaled by multiplying `E-3` for means and standard deviations.

| #      | 15            | 25            | 50            | 100           |
|--------|---------------|---------------|---------------|---------------|
| ANP    | 3.0($\pm$0.36) | 2.8($\pm$0.41) | 2.5($\pm$0.14) | 2.8($\pm$0.19) |
| MoE-NP | **2.0($\pm$0.45)** | 1.9($\pm$0.28) | **1.4($\pm$0.06)** | **1.5($\pm$0.06)** |

### I.5 Augmenting MoE-NPs with Convolutional Modules

In this section, we examine the chance of Since neural architectures of encoders are quite different between ConvCNPs [17] and previous mentioned NP baselines, we only report the results of NP related models with the same functional encoder structures in the main paper. Note that the translation equivariance is injected in ConvCNPs, which is a strong inductive bias for image dataset. Naturally,

Table 6: Pixel-wise mean squared errors (MSEs) in the image completion tasks on the CIFAR10 dataset. The number of random context points is varied in a range $(10, 200, 500, 800, 1000)$ to test performance at different levels.

| # | 10 | 200 | 500 | 800 | 1000 |
|---|---|---|---|---|---|
| ANP | 0.0377 | 0.0223 | 0.0217 | 0.0215 | 0.0215 |
| MoE-NP | **0.0377** | **0.0142** | **0.0117** | **0.0110** | **0.0107** |

we also develop the convolutional version of MoE-NPs. And we report the separate results in image completion here. In our settings, MoE-ConvCNPs use 3 experts in convolutional modules. The implementation of ConvCNPs can be found in [17]. It can be seen that in Table. (7), in comparison to ConvCNP, the image completion performance is further improved with help of multiple experts.

Table 7: Pixel-wise mean squared errors (MSEs) in the image completion tasks on the CIFAR10 dataset. The number of random context points is varied in a range $(10, 200, 500, 800, 1000)$ to test performance at different levels.

| # | 10 | 200 | 500 | 800 | 1000 |
|---|---|---|---|---|---|
| ConvCNPs | 0.036 | 0.0062 | 0.002 | 0.0011 | 0.0019 |
| MoE-ConvCNPs | **0.035** | **0.0057** | **0.0017** | **0.0007** | **0.0009** |

### I.6 More Visualization Results

Here we include more visualized CelebA images by varying the number of observed pixels. These are produced using CNN Augmented MoE-NPs (MoE-ConvCNPs). Fig.s (14)/(15)/(16)/(17) are image completion results given the fixed number of random context pixels. Fig.s (18)/(19)/(20)/(21) are image completion results given the fixed number of ordered context pixels.

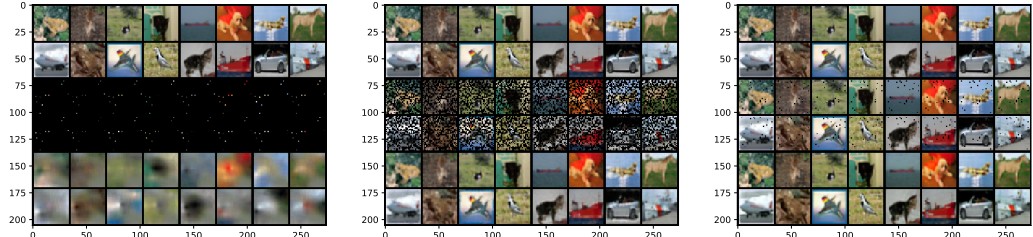
Figure 14: Image Completion Results using CNN Augmented MoE-NPs.

## J  Computational Devices

Throughout the research process, we use NVIDIA 1080-Ti GPUs and Pytorch is used as the deep learning toolkit.

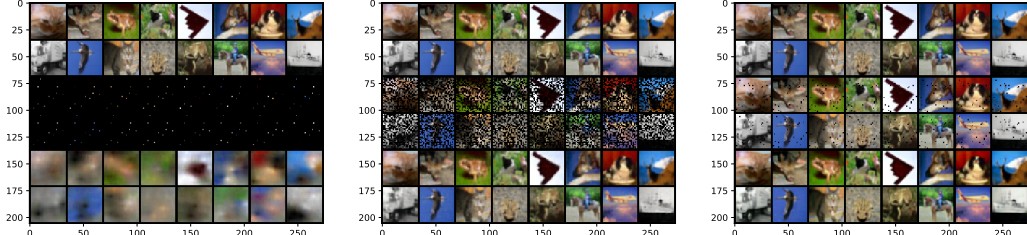

Figure 15: Image Completion Results using CNN Augmented MoE-NPs.

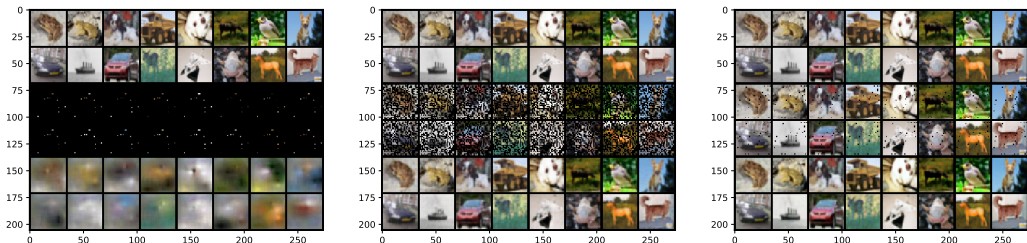

Figure 16: Image Completion Results using CNN Augmented MoE-NPs.

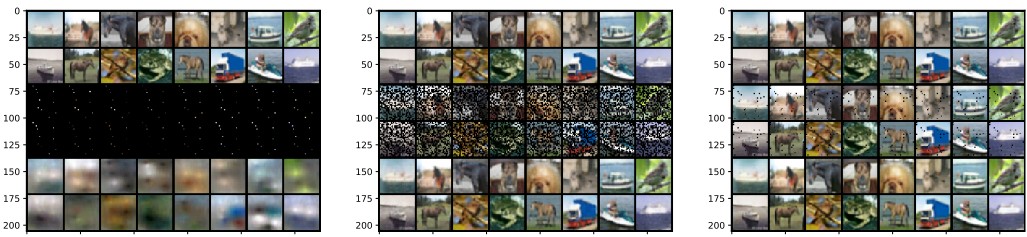

Figure 17: Image Completion Results using CNN Augmented MoE-NPs.

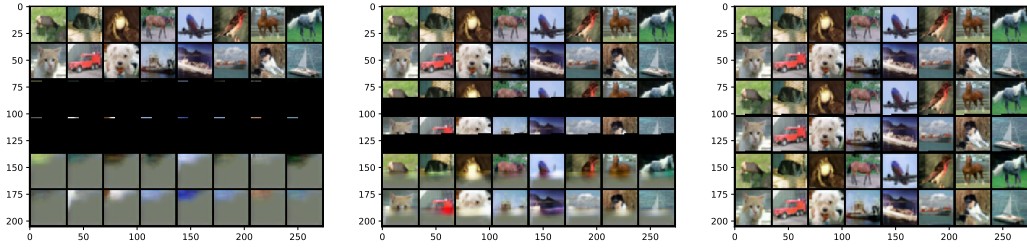

Figure 18: Image Completion Results using CNN Augmented MoE-NPs.

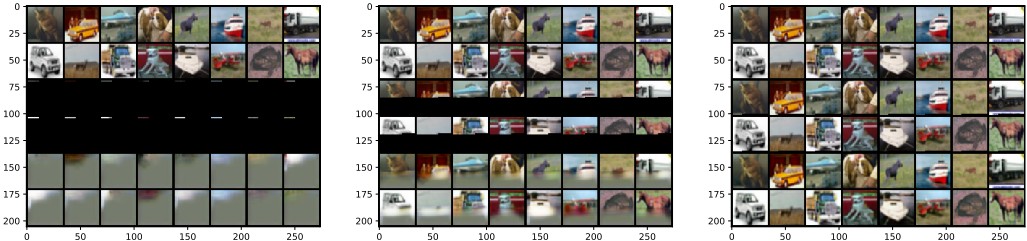

Figure 19: Image Completion Results using CNN Augmented MoE-NPs.

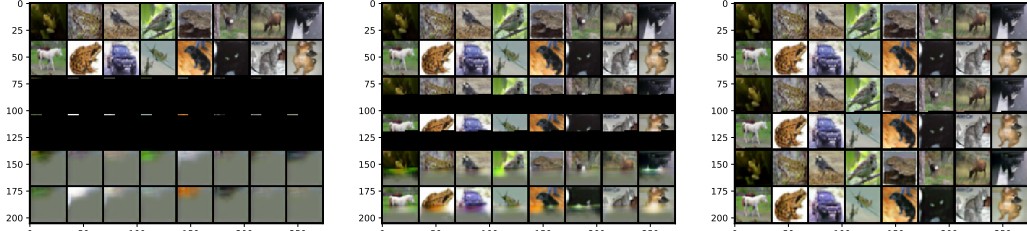

Figure 20: Image Completion Results using CNN Augmented MoE-NPs.

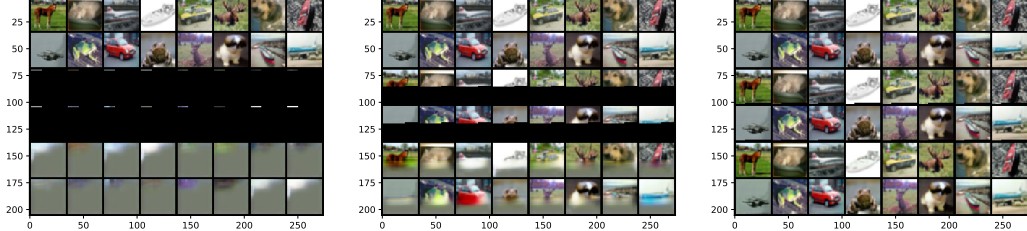

Figure 21: Image Completion Results using CNN Augmented MoE-NPs.