# OpenReview forum: "Learning Expressive Meta-Representations with Mixture of Expert Neural Processes"
_NeurIPS.cc/2022/Conference — NeurIPS 2022 Accept_

### Official Review · Reviewer_fZgG · 2022-07-08

**Rating:** 6
**Confidence:** 4
**Soundness:** 3 good
**Presentation:** 2 fair
**Contribution:** 3 good

**Summary:**

This paper proposes a new Neural Processes family, Mixture of Expert Neural Processes (MoE-NPs). While NPs represent each task as a single vector representation sampled from Gaussian distribution, MoE-NPs use discrete representation to select which representation will be used for each data. Therefore, MoE-NPs have more flexibility than other NPs and it is shown in this paper empirically from 1D sinusoidal regression task to Meta Reinforcement Learning tasks.


==========================
The authors addressed my concerns enough, I increase my score.

**Questions:**

- In Table 4, ANPs results are same for the different number of given contexts. Is it right? or typo?
- As described in the above session, I am not sure when MoE-NPs is strong. When we need to use different $z$ in a task?
- In section 5.2, the amplitudes or phases of $sin$ and $cos$ function, is it right?


**Limitations:**

I think the authors already addressed the limitations of their model adequately, but more clarifying why it is required for some cases will make this paper more concrete.

**Strengths And Weaknesses:**

Strengths:
- MoE-NPs take the idea of Mixture of Expert to NPs and show better expressiveness than other NPs.
- Equations are derived in appendix, which helps to understand.
- Including ANP and Conv-NP, they evaluated MoE-NPs with a wide range of baselines comprehensively.

Weaknesses
- I can understand MoE-NPs can have more representative power than NPs, but I am not sure when we need it. The toy regression task is to show the strength of this method, but except that, I cannot find the killer applications for MoE-NPs. MoE-NPs shows more stable results than others for image completion in Figure 4, but the y-axis range is quite small, so I am not sure we can say MoE-NPs is necessary for that. For RL, MoE-NP didn't show a significant performance gap from baselines.
- In Figure 8, it is also weakness that overestimated experts hurt the performance. I think it is because by representing a single underlying function with more than 1 expert, the model misses the underlying patterns.
- In section 4.2, notations are quite confusing (e.g., $q_{\pi 2,1}$ and $p_{\pi 2,2}$ are the variational posterior and prior for categorical variable, why $(2,1)$ and $(2,2)$?). They are not well described before this section, so it is hard to understand.
- For toy regression task (section 5.2), two functions look no variations without the noise. For example, in [1], they made variations on amplitudes or phase, but as described in section 5.2, this task doesn't have those variations to learn the meta-learning framework. I don't think that it is enough to validate those meta-learning models.

[1] Finn, Chelsea, Pieter Abbeel, and Sergey Levine. "Model-agnostic meta-learning for fast adaptation of deep networks." International conference on machine learning. PMLR, 2017.

---

> ### Author Response · Authors · 2022-08-01
> **Answer to Reviewer fZgG**
>
> We sincerely express our gratitude for your comments in helping improve our manuscript. Responses to each point are as follows.
>
>
> ***1. Scalability with More Expert Latent Variables***
> This question is similar to that posed by Reviewer ZLhe. Due to the response character numbers limit, please refer to the **Answer to Reviewer ZLhe**. Note that we updated our submission accordingly and you can find added content there. **Line number** in bold refers to the added contents.
>
>
> ***2. Applications and Necessity of MoEs***
>
> Thanks for this question. Image completion is a challenging task in computer vision domains and the range of completion error for SOTA models is on a scale of E-2 or E-3 (Refer to that in CAVIA). Though the difference is tiny on a value scale, the visual result can be quite different. Taking a look at Fig.5, you can find the image is blurred a lot with fewer pixels though the scale of completion errors is tiny. So it is necessary to obtain a better result in this domain.
>
> Our method is a generalization of NPs. So it can address the tasks suitable for NPs family. The inductive bias of multiple experts makes it more appropriate for the case when a task is dominated by multiple factors in latent functional space. The following part can be found in the updated manuscript **Line639-646**
>
> Here we provide two available applications with MoE-NPs in the industry. One is in multilingual machine translation or multilingual language auto-completion as mentioned by **Reviewer et6m**. In this case, a mixture of experts corresponds to multilingual functional priors for multi-modal signals [1] and enables the prediction with partial observations. Another application lies in modeling irregular time series [2-3]. In this case, diverse experts can handle discontinuous components in a rich family of stochastic functions. Meanwhile, the entropy of learned assignment latent variables can tell us the regions likely to be discontinuous, which is quite helpful in anomaly detection in a black-box system.
>
> *Reference*:
>
> [1] Shi, Yuge, Brooks Paige, and Philip Torr. "Variational mixture-of-experts autoencoders for multi-modal deep generative models." Advances in Neural Information Processing Systems 32 (2019).
>
> [2] Rubanova, Yulia, Ricky TQ Chen, and David K. Duvenaud. "Latent ordinary differential equations for irregularly-sampled time series." Advances in neural information processing systems 32 (2019).
>
> [2] Schirmer, Mona, et al. "Modeling irregular time series with continuous recurrent units." International Conference on Machine Learning. PMLR, 2022.
>
>
> ***3. Complicated Notations of Variational Distributions***
>
> Sorry for the complicated notations. We agree with you and will move a more detailed introduction about these distributions to the Sec.4 in the final manuscript if one additional page is allowed.
>
>
> ***4. More Toy Regression Results***
>
> You are right. We generate diverse functions by simply injecting different random noise into the output. The aim of this toy experiment is to understand the meaning of expert latent variables and assignment latent variables.
> To address your concern, we’ve added more experiments by varying the amplitudes in the interval $[0.1,5.0]$ or phases in the interval $[0.0, \pi]$ in toy experiments. Please refer to **Line972-986**.  The meta-testing results are reported in **Line985-Table4**. We copied the mean square errors of 500 sampled testing tasks as follows. It shows ANP achieves the best performance, followed by MoE-NPs.
>
> | CNP         | NP          | FCRL      | ANP         | MoE-NP      |
> |-------------|-------------|-----------|-------------|-------------|
> | 0.053(1E-4) | 0.070(3E-4) | 0.04(0.0) | 0.027(2E-4) | 0.032(1E-4) |
>
>
>
> ***5. Others***
>
> You are right. It was a typo in the former Table4 and thank you for pointing it out. Meanwhile, we’ve proofread the manuscript a couple of times during the rebuttal phase, included new results, and revised most of the typos or irregular notations. We’ll keep revising the manuscript according to your precious feedback during this period.
>
> Finally, many thanks for your comment and your suggestions are indeed helpful in improving our manuscript.

---

> > ### Comment · Reviewer_fZgG · 2022-08-05
> > **Appreciate your informative response**
> >
> > Hi authors,
> >
> > thank you for your great response.
> >
> > I think that you covered almost all my concerns, specially, new results for scalability with more expert variables is impressive. Thanks!

---

### Official Review · Reviewer_ZLhe · 2022-07-10

**Rating:** 7
**Confidence:** 3
**Soundness:** 4 excellent
**Presentation:** 4 excellent
**Contribution:** 4 excellent

**Summary:**

This paper studies the problem of learning-to-learn a new task $\tau$ from a dataset $D_{\tau}$ of labeled pairs $(x_1, y_1), ..., (x_n, y_n)$. The authors propose to do so using meta learning with neural processes (NPs), trained via variational inference. They propose a new type of NP model that uses a multi-modal mixture density as a prior, called Mixture of Expert Neural Processes (MoE-NPs).

The problem with vanilla NPs is that they use a single gaussian latent variable z to specify an entire task. As result they have limited expressiveness when trying to learn functional priors over tasks from unknown distributions. The authors propose to resolve this issue by using a mixture prior with K gaussian distributions and latent variables $z_{1:K}, e$ where $e$ is a categorical mixture assignment variable.

**Questions:**

There is an important choice that has to be made for the MoE-NP model, which is the number of mixtures K. In all of the experiments the authors take this number as a given (K=3 for Acrobot, K=2 for CIFAR-10), but I assume there is a lot of tuning and optimization that goes on behind the scenes to settle upon a value that works well. I would suggest that the authors say a little bit about this tuning procedure and what it looked like for the experiments.

Pretty much all of the implementation details of the model are left to the appendix, which is non-standard. I suggest finding some room in the main paper to say at least a little bit about the implementation you're working with.

**Limitations:**

Limitations are adequately addressed

**Strengths And Weaknesses:**

The paper writing is generally clear and thorough. Although I was unfamiliar with NPs before reading the paper, I believe that the proposed extension provides an original and significant contribution to the field based on the provided theory and experiments.

---

> ### Author Response · Authors · 2022-08-01
> **Answer to Reviewer ZLhe**
>
> We sincerely express our gratitude for your comments in helping improve our manuscript. Responses to each point are as follows.
>
> ***1. Scalability with More Expert Latent Variables***
> This question is similar to that posed by Reviewer et6m. We’ve copied the answer here. Note that we updated our submission accordingly and you can find added content there. **Line number** in bold refers to the added contents.
>
> *(Phenomenon)* In experiments, we observe by increasing the number of experts to a certain level, e.g. in the Acrobot system >5 cases, the performance of MoE-NP cannot be improved and even deteriorates.
>
> *(Analysis)* Here we attribute this to inference sub-optimality. Since the developed MoE-NP is a VAE-like model, the expert latent variables’ number $K$ decides $e$’s dimension and the encoder of variational posterior is $q_{\phi_{2,1}}(e\vert x,y,z_{1:K})$ with $y$ as the input. In auto-encoder models, when the dimension of latent variables is higher than that of the input, the model tends to copy the input to the output and fails to learn representations. This is the direct source of overfitting and applies to conditional VAE methods. Remember that $y$ dimensions in Acrobot and CIFAR10 are respectively 6 and 3.
>
> *(Theoretical Solution)* Indeed, we have another implementation to avoid overfitting. That is to change the inference way for $e$. Here we updated this part in **Line793-828**, in which we set the variational posterior without y information, namely $q_{\phi_{2,1}}(e\vert x,y,z_{1:K})=p_{\phi_{2,2}}(e\vert x,z_{1:K})$. You can find the gradient estimator for $e$ in **Line862-868** as follows.
>
>    $$\frac{\partial}{\partial\phi_{2,2}}L(y;x,\theta,\phi_1,\phi_2)=E_{q_{\phi_1}(z_{1:K}\vert D_{\tau}^{T})}[\sum_{k=1}^{K}[\frac{\partial}{\partial\phi_{2,2}}p_{\phi_{2,2}}(e_{k}=1\vert x,z_{1:K})]\ln p_{\theta}(y\vert x,z_k)]$$
>         $$=E_{q_{\phi_1}(z_{1:K}\vert D_{\tau}^{T})}[\sum_{k=1}^{K}[p_{\phi_{2,2}}(e_{k}=1\vert x,z_{1:K})\frac{\partial}{\partial\phi_{2,2}}\ln p_{\phi_{2,2}}(e_{k}=1\vert x,z_{1:K})]\ln p_{\theta}(y\vert x,z_k)]$$
>
> *(Summaries)* When applying the approximate posterior with y as the input, better performance can be achieved by setting the optimal expert number no greater than y’s dimension. When setting the approximate posterior as $q_{\phi_{2,1}}(e\vert x,y,z_{1:K})=p_{\phi_{2,2}}(e\vert x,z_{1:K})$, we do not need to worry about overfitting problems.
> We added new results/discussions in CIFAR10 image completion in **Line964-Table3**, which shows increasing the number of experts can significantly improve performance when setting $q_{\phi_{2,1}}(e\vert x,y,z_{1:K})=p_{\phi_{2,2}}(e\vert x,z_{1:K})$.
> Shown in the Table is the image completion result with varying number of context points when $q_{\phi_{2,1}}(e\vert x,y,z_{1:K})=p_{\phi_{2,2}}(e\vert x,z_{1:K})$.
>
> | #                   | 10     | 200    | 500    | 800    | 1000   |
> |---------------------|--------|--------|--------|--------|--------|
> | MoE-NPs (3 experts) | 0.0482 | 0.0183 | 0.0170 | 0.0166 | 0.0165 |
> | MoE-NPs (5 experts) | 0.0362 | 0.0103 | 0.0171 | 0.0061 | 0.0057 |
> | MoE-NPs (7 experts) | 0.0359 | 0.0095 | 0.0062 | 0.0052 | 0.0048 |
>
>
> ***2. Tuning Procedure***
>
> This is a good question.
> In practice, we didn’t tune other hyperparameters (except the number of experts) or use any non-trivial tricks in training MoE-NPs. However, we suggest the number of experts no greater than the dimension of the output $y$. Another important trick is to use the approximate posterior of discrete variables in the form $q_{\phi_{2,1}}(e\vert x,y,z_{1:K})=p_{\phi_{2,2}}(e\vert x,z_{1:K})$ can bring better performance and this does not suffer overfitting issues.
>
>
> ***3. Moving Contents to Main Paper***
>
> Sorry for our organization in content. We agree with you. And more implementation details will be moved to the main paper if one additional page is allowed.
>
>
> Finally, many thanks for your precious advice, which indeed improves our manuscript a lot.

---

### Official Review · Reviewer_et6m · 2022-07-11

**Rating:** 7
**Confidence:** 3
**Soundness:** 3 good
**Presentation:** 3 good
**Contribution:** 3 good

**Summary:**

The paper has a very good motivation of combining neural processes (NP) and a mixture of experts (MoE) with the hope to further strengthen the power of NP as MoE usually does with other architectures in the application of context-based meta-learning. To enable a collection of experts, a single global latent variable is replaced by a collection of latent variables, which represent MoE NPs.

**Questions:**

Line 66/67 seems to be overly stated, implying gradient-based optimization is slower than NP, which might not be true. Gaussian Processes have cubic cost without proper modification, and NP marries GP and NN to inherit the fast training. Consequently, it is not clear whether NP is strictly faster than NN, at least based on this hybrid intuition. Please elaborate or explain why your claim is so.


It seems for simpler tasks, the model performs really well but with more difficult tasks, the variance is much higher (Fig 7), is it because of variational and Monte Carlo approximations over so many latent variables?


Similar to the scalability point in Weakness above: although the authors already presented a high performance in difficult settings, e.g. Humannoid, it would be more practical for this solution if you could also demonstrate a higher-scale experiment in which there are a big number of tasks, maybe meta-learning for multilingual MT.


See also in Weakness above.


**Ethics Review Area:**

["I don’t know"]

**Strengths And Weaknesses:**

Strength
1. Apart from motivation, the paper has clear explanations of step-by-step derivations.
2. Experiments show strong improvements over the chosen baselines.

Weakness
1. One important question in practice is scalability. Naturally expanded from NP, it would be good to compare it with at least NP since the model is more and more complex as more latent variables are injected. Also, it’s not clear (as also mentioned in 5.5) why an increasing number of experts would not help after 5?

- So, I would suggest doing some expert-specific studies about mean/variances, learned representation, and system-related statistics to see the difference between them. One hypothesis is that the model collapses somewhere that only a few experts actually learn and thus increasing the number of experts would not help. This can vary by tasks/data distributions as well.

- Similarly for Latent Variables ablation in 5.5, I would suggest doing a per-task, per-expert analysis to see if it matches the intuition on whether each expert is specialized for each (group of) task(s), which would further solidify the intuition to replace a global variable with different local variables.

2. Despite a very good motivation and setting, there might be a need to compare with some other gradient-based MoE performance because MoE naturally enhances models’ complexity (as in the same case here with the introduction of more latent variables) for more power, in exchange for being more complex and other costs such as communication and/or memory. Likewise, it would be good if we also have some comparison with some other MoE baselines such as GShard, Deepspeed MoE, …

3. No code is provided.

---

> ### Author Response · Authors · 2022-08-01
> **Answer to Reviewer et6m**
>
> We sincerely express our gratitude for your comments in helping improve our manuscript. Responses to each point are as follows.
>
> ***1. Scalability with More Expert Latent Variables***
>
> *(Phenomenon)* In experiments, we observe by increasing the number of experts to a certain level, e.g. in the Acrobot system >5 cases, the performance of MoE-NP cannot be improved and even deteriorates.
>
> *(Analysis)* Here we attribute this to inference sub-optimality. Since the developed MoE-NP is a VAE-like model, the expert latent variables’ number $K$ decides $e$’s dimension and the encoder of variational posterior is $q_{\phi_{2,1}}(e\vert x,y,z_{1:K})$ with $y$ as the input. In auto-encoder models, when the dimension of latent variables is higher than that of the input, the model tends to copy the input to the output and fails to learn representations. This is the direct source of overfitting and applies to conditional VAE methods.
>
> *(Theoretical Solution)* Indeed, we have another implementation to avoid overfitting. That is to change the inference way for $e$. Here we updated this part in **Line793-828**, in which we set the variational posterior without y information, namely $q_{\phi_{2,1}}(e\vert x,y,z_{1:K})=p_{\phi_{2,2}}(e\vert x,z_{1:K})$. You can find the gradient estimator for $e$ in **Line862-868** as follows.
>
>    $$\frac{\partial}{\partial\phi_{2,2}}L(y;x,\theta,\phi_1,\phi_2)=E_{q_{\phi_1}(z_{1:K}\vert D_{\tau}^{T})}[\sum_{k=1}^{K}[\frac{\partial}{\partial\phi_{2,2}}p_{\phi_{2,2}}(e_{k}=1\vert x,z_{1:K})]\ln p_{\theta}(y\vert x,z_k)]$$
>         $$=E_{q_{\phi_1}(z_{1:K}\vert D_{\tau}^{T})}[\sum_{k=1}^{K}[p_{\phi_{2,2}}(e_{k}=1\vert x,z_{1:K})\frac{\partial}{\partial\phi_{2,2}}\ln p_{\phi_{2,2}}(e_{k}=1\vert x,z_{1:K})]\ln p_{\theta}(y\vert x,z_k)]$$
>
> *(Summaries)* When applying the approximate posterior with y as the input, better performance can be achieved by setting the optimal expert number no greater than y’s dimension. When setting the approximate posterior as $q_{\phi_{2,1}}(e\vert x,y,z_{1:K})=p_{\phi_{2,2}}(e\vert x,z_{1:K})$, we do not need to worry about overfitting problems.
> We added new results/discussions in CIFAR10 image completion in **Line964-Table3**, which shows increasing the number of experts can significantly improve performance when setting $q_{\phi_{2,1}}(e\vert x,y,z_{1:K})=p_{\phi_{2,2}}(e\vert x,z_{1:K})$.
> Shown in the Table is the image completion result with varying number of context points when $q_{\phi_{2,1}}(e\vert x,y,z_{1:K})=p_{\phi_{2,2}}(e\vert x,z_{1:K})$.
>
> | #                   | 10     | 200    | 500    | 800    | 1000   |
> |---------------------|--------|--------|--------|--------|--------|
> | MoE-NPs (3 experts) | 0.0482 | 0.0183 | 0.0170 | 0.0166 | 0.0165 |
> | MoE-NPs (5 experts) | 0.0362 | 0.0103 | 0.0171 | 0.0061 | 0.0057 |
> | MoE-NPs (7 experts) | 0.0359 | 0.0095 | 0.0062 | 0.0052 | 0.0048 |
>
> ***2. Examine Latent Variables with Statistics***
> You are right. We take your advice. Since it is difficult to examine in complicated cases (varying number of context points, high dimensional latent variables), we propose to answer this in 1-d toy regression. We also checked the learned expert latent variables and didn’t find the collapse issue. But the entropy of discrete variables can explain more.
>
> We computed the entropy values of the learned conditional prior and added results/analysis in **Line962-Fig12**.
> $$ H[e]=\sum_{k=1}^{K}-p_{\phi_{2,2}}(e_{k}=1\vert z_{1:K},x_i)\ln p_{\phi_{2,2}}(e_{k}=1\vert z_{1:K},x_i)
>         =\sum_{k=1}^{K}-\alpha_k\ln\alpha_k
> $$
>
> ***3. Other MoE Methods***
> Thanks for your suggestion. We haven’t found other implementations of MoE with meta-learning probabilistic models. So the simplified MoE module is used here to enable learning of diverse functional priors. We added this discussion and MoE references in Appendix **Line639-646**.
>
> ***4. Other Questions***
>
> **Over-stated Claims in Line66-67**. We agree with you. Here we want to express the trait of NPs’ the predictive distribution $p(y_*\vert x_*,[x_c,y_c])$. It does not require computing matrix inversion, e.g. Kernel matrix in GPs, or gradient updates in prediction.  This does not mean NPs are faster than gradient-based ones. To avoid ambiguity, we rewrite this claim in **Line66-67**.
>
> **Comparison with Complicated NPs**. We agree with you. We’ve included more complicated NPs, such as ANPs, to compare with ours in Appendix I.4.
>
> **Available Extension and Codes**. We agree with you. Our method has the potential to deploy in multilingual machine translation or tasks multi-signals and we added your suggestion in **Line639-646**. Meanwhile, you can find our anonymous code in **Line847-852**.
>
> **Variance Observation**. In more difficult tasks, e.g. Humanoid, there are multiple sources for the variances: (i) stochastic gradient estimator (ii) complexity of the dynamical system and state distributions.
>
> Thanks again for your suggestions, we hope your concerns are well addressed in this phase.

---

> > ### Comment · Reviewer_et6m · 2022-08-08
> > **Rebuttal**
> >
> > I have read the questions and answers for all reviewers and found the authors have made an extra effort to address each question/comment in a short period of time. For this reason, I am increasing 1 point to my score and thank the authors for the rebuttals.

---

### Author Response · Authors · 2022-08-09
**Reply to Post Rebuttal**

Dear All Reviewers and Area Chairs,

Thank you all for your comments and constructive suggestions for our manuscript. Your engagement in the review/rebuttal period helps improve our manuscript a lot. Finally, we express our gratitude for your efforts in helpful reviews and discussions.

---

### Meta-Review · Area_Chair_Etga · 2022-08-27

**Recommendation:** Accept
**Confidence:** Certain

**Metareview:**

The paper introduces a mixture of expert prior in the neural processes (NP) models as a way of improving the expressibility of the prior and posterior. The paper is well motivated, written well, and the method is reasonable and sound. The experiment results are also comprehensive and convincing. The authors well addressed the reviewer's questions in the rebuttal. All reviewers agreed on accepting this paper.

**Award:**

No

---

### Decision · Program_Chairs · 2022-09-14

Accept